# Feedback Schrödinger Bridge Matching

**Panagiotis Theodoropoulos**[1], **Nikos Komianos**[1], **Vincent Pacelli**[1],
**Guan-Horng Liu**[2†], **Evangelos A. Theodorou**[1†]
[1]Georgia Institute of Technology  [2]FAIR, Meta
{ptheodor3,nkomianos3,vpacelli3,evangelos.theodorou}@gatech.edu
ghliu@meta.com

## Abstract

Recent advancements in diffusion bridges for distribution transport problems have heavily relied on matching frameworks, yet existing methods often face a trade-off between scalability and access to optimal pairings during training. Fully unsupervised methods make minimal assumptions but incur high computational costs, limiting their practicality. On the other hand, imposing full supervision of the matching process with optimal pairings improves scalability, however, it can be infeasible in most applications. To strike a balance between scalability and minimal supervision, we introduce **Feedback Schrödinger Bridge Matching (FSBM)**, a novel *semi-supervised* matching framework that incorporates a small portion ($< 8\%$ of the entire dataset) of pre-aligned pairs as *state feedback* to guide the transport map of non-coupled samples, thereby significantly improving efficiency. This is achieved by formulating a static Entropic Optimal Transport (EOT) problem with an additional term capturing the semi-supervised guidance. The generalized EOT objective is then recast into a dynamic formulation to leverage the scalability of matching frameworks. Extensive experiments demonstrate that FSBM accelerates training and enhances generalization by leveraging coupled pairs' guidance, opening new avenues for training matching frameworks with partially aligned datasets.

## 1 Introduction

Transporting samples between distributions is a ubiquitous problem in machine learning. Given the rise of generative modeling, significant progress has been made using static (Goodfellow et al., 2014), deterministic (Chen et al., 2018; Biloš et al., 2021), or stochastic mappings (Ho et al., 2020; Song et al., 2020) that transport samples from noise to a more complex distribution. A prominent approach lies in diffusion models that simulate a Stochastic Differential Equation (SDE) to diffuse the data to noise, and learn the score function (Hyvärinen & Dayan, 2005; Vincent, 2011) to reverse the process (Anderson, 1982). However, these models present several limitations. For instance, the requirement to converge during the noising (forward) process to a Gaussian noise suggests that these models must run for a sufficient number of time steps to ensure the final distribution approximates Gaussian noise (Chen et al., 2021). Additionally, this suggests that these models begin their generative (backward) processes without any structural information about the data distribution, which implies randomness being introduced in the couplings that emerge from the diffusion models (Liu et al., 2023). Lastly, there is no guarantee that the optimal path interpolating the boundary distribution minimizes the kinetic energy. (Shi et al., 2023)

In an attempt to overcome these shortcomings, principled approaches that stem from Optimal Transport (Villani et al., 2009) have emerged. The most prominent alternative has been the Schrödinger Bridge (SB; Schrödinger (1931)), which has been shown to be equivalent to entropy-regularized Optimal Transport (EOT; Cuturi (2013); Léonard (2013); Pavon et al. (2021)), and can also be framed as a Stochastic Optimal Control (SOC) Problem (Chen et al., 2016; 2021). In particular, SB gained significant popularity in the realm of generative modeling following advancements proposing a training scheme based on the Iterative Proportional Fitting (IPF), a continuous state space extension of the Sinkhorn algorithm to solve the dynamic SB problem (De Bortoli et al., 2021; Vargas et al., 2021). Notably, SB generalizes standard diffusion models transporting data between arbitrary distributions

---

[†]Equal advising. Work done while Guan was at Georgia Tech.

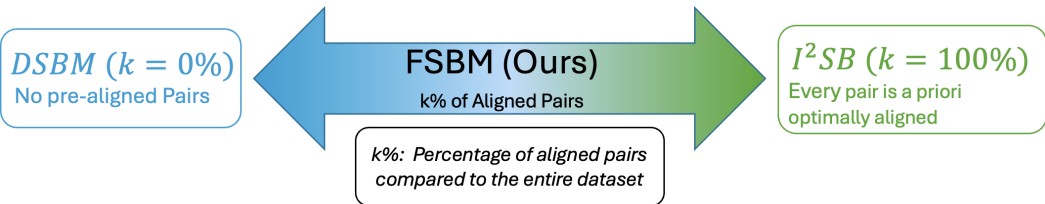

Figure 1: FSBM connecting existing Bridge Matching frameworks at the extremes, where the dataset is comprised of either fully aligned or fully non-aligned pairs

$\pi_0, \pi_1$ with fully nonlinear stochastic processes, seeking the unique path measure that minimizes the kinetic energy. More recently, building on advancements in Bridge Matching methods (Peluchetti, 2023; Liu et al., 2022c), Shi et al. (2023) introduced Diffusion Schrödinger Bridge Matching (DSBM), a framework that solves the Schrödinger Bridge problem while being significantly more efficient. Unlike previous methods, DSBM avoids the need to cache the full trajectories of forward and backward SDEs, making it more scalable and mitigating the time-discretization and "forgetting" issues encountered in earlier DSB techniques. For training these matching frameworks, we start from two datasets that lack pre-aligned pairs—that is, we sample only from the marginal $\pi_0, \pi_1$—and the framework is trained to determine the optimal coupling in the sense of least-energy transportation. This process represents an unsupervised learning approach, as no prior information about the coupling is provided. However, there are other applications—such as image restoration (Liu et al., 2023) and protein docking (Somnath et al., 2023)—in which we have prior knowledge of the coupling $\pi_{0,1}$, between pairs $(x_0, x_1) \sim \pi_{0,1}$ drawn presumably from the SB solutions. Consequently, using an unsupervised approach for such problems is not ideal, as it discards valuable information about the relationships between the samples of the two boundary distributions. An alternative approach suggests reframing these tasks as inverse problems. In this vein, Liu et al. (2023) proposed recently $I^2SB$, building diffusion bridge matching frameworks between coupled data to perform image restoration.

In practice, in most tasks, the available datasets might possess a few limited coupled pairs, however, the high cost of manually labeling large datasets renders fully supervised approaches infeasible. Unfortunately, existing matching frameworks can not effectively leverage the information in partially pre-aligned datasets. An alternative approach from Optimal Transport (OT) literature involves using guided transportation maps, where only a few pairs that belong to a Key-Point (KP) set are annotated. These pairs are utilized to guide the transportation mapping of the unpaired samples (Gu et al., 2023a), substantially reducing the need for extensive human labeling or expert guidance by leveraging a small number of source-target aligned sample pairs for training (Mustafa & Mantiuk, 2020). More specifically, recent approaches employ a "relation-preserving" scheme, which maintains the data's relationship to the given pairs from the KP set (Mémoli, 2011; Sato et al., 2020), or a pairwise distance-preserving constraint (Gu et al., 2023b). However, the feasibility of adapting semi-supervised guidance in a dynamic Schrödinger Bridge or Bridge Matching setting remains an open question. In this vein, drawing inspiration from the guided OT schemes, we advocate a semi-supervised guided Schrödinger Bridge Matching framework.

In this work, we introduce a novel semi-supervised matching algorithm, **Feedback Schrödinger Bridge Matching (FSBM)**, designed to integrate information from partially aligned datasets. Drawing inspiration from optimal transport (OT) literature (Gu et al., 2022), our analysis begins from a static, semi-supervised OT problem, from which we derive a dynamic objective. Following recent advancements in matching frameworks, we adopt an alternating scheme, where the intermediate path and the coupling are optimized in two separate steps, resulting in a novel matching framework that leverages partially aligned data to guide the transport mapping of non-aligned samples. A key aspect of our approach is that the information from aligned samples is encoded as **state feedback** within the dynamic objective, effectively steering the transport of non-aligned data. This renders our FSBM a bridge between two extremes: unsupervised matching frameworks, which lack pre-aligned couplings (Shi et al., 2023), and fully supervised frameworks, where data is entirely pre-aligned (Liu et al., 2023; Somnath et al., 2023) (see Figure 1). Empirical results show that our algorithm generalizes better, is more robust to perturbations in the initial conditions, and exhibits reduced training time. Our contributions are summarized as follows:

- We introduce FSBM, the first matching framework leveraging information from partially aligned datasets.

- We begin our analysis from a static semi-supervised OT problem, and derive a modified dynamic formulation. Notably, our analysis can be extended to any selection of regularization functions.

- We introduce an Entropic Lagrangian extension of the variational gap objective used in BM frameworks to match the parameterized drift given a prescribed probability path

- Through extensive experimentation, we verify the remarkable capability of FSBM to generalize under a variety of unseen initial conditions, while simultaneously achieving faster training times. These results were consistent across a wide range of matching tasks from low-dimension crowd navigation, to high-dimensional opinion depolarization and image translation.

## 2 PRELIMINARIES

### 2.1 SCHRÖDINGER BRIDGES

The Schrödinger Bridge (Schrödinger, 1931) in the path measure sense is concerned with finding the optimal measure $\mathbb{P}^\star$ that minimizes the following optimization problem

$$\min_{\mathbb{P}} KL(\mathbb{P}|\mathbb{Q}), \quad \mathbb{P}_0 = \pi_0, \ \mathbb{P}_1 = \pi_1 \tag{1}$$

where $\mathbb{Q}$ is a Markovian reference measure. Hence the solution of the dynamic SB $\mathbb{P}^\star$ is considered to be the closest path measure to $\mathbb{Q}$. A very convenient, and heavily used property of $\mathbb{P}^\star$ is its decomposition as $P^\star = \int_{\mathbb{R}^d \times \mathbb{R}^d} \mathbb{Q}_{|0,1} d\pi^\star(x_0, x_1)$, where $\pi^\star$ is the solution of the static SB $\min_{\pi \in \Pi(\pi_0, \pi_1)} KL(\pi|R)$, and $Q_{|0,1}$ is the Bridge of the reference measure for pinned points $(x_0, x_1)$ (Léonard, 2013). Another formulation of the dynamic SB crucially emerges by applying the Girsanov theorem in Eq. (1)

$$\min_{u_t, p_t} \int_0^1 \mathbb{E}_{p_t}[\|u_t\|^2]dt \quad \text{s.t.} \ \frac{\partial p_t}{\partial t} = -\nabla \cdot (u_t p_t) + \frac{\sigma^2}{2}\Delta p_t, \quad \text{and} \quad p_0 = \pi_0, \quad p_1 = \pi_1 \tag{2}$$

Finally, note that the static SB is equivalent to the entropy regularized OT formulation (Pavon et al., 2021; Nutz, 2021).

$$\min_{\pi \in \Pi(\pi_0, \pi_1)} \int_{\mathbb{R}^d \times \mathbb{R}^d} \|X_0 - X_1\|^2 d\pi(X_0, X_1) + \epsilon KL(\pi|\pi_0 \otimes \pi_1) \tag{3}$$

This regularization term enabled efficient solution through the Sinkhorn algorithm and has presented numerous benefits, such as smoothness, and other statistical properties (Ghosal et al., 2022; Léger, 2021; Peyré et al., 2019).

### 2.2 SCHRÖDINGER BRIDGE MATCHING

As discussed above, SB Matching algorithms (Shi et al., 2023; Liu et al., 2024) have emerged recently, following advancements in matching frameworks (Liu et al., 2022c; Peluchetti, 2023; Lipman et al., 2022; Liu et al., 2022b), rendering the solution of the SB problem significantly more tractable. Specifically, this family of algorithms solves Eq. (2) by separating the training process into two alternating steps. The first step entails relaxing the boundary distributional constraints in Eq. (2) into just two endpoints, by fixing the coupling $\pi_{0,1}$, and drawing pairs of samples $(x_0, x_1)$.

$$\min_{u_{t|0,1}} \int_0^1 \mathbb{E}_{p_{t|0,1}}[\|u_{t|0,1}\|^2]dt \tag{4}$$
$$\text{s.t.} \ \frac{\partial p_{t|0,1}}{\partial t} = -\nabla \cdot (p_{t|0,1} u_t) + \frac{1}{2}\sigma^2 \Delta p_{t|0,1}, \quad X_0 = x_0, \quad X_1 = x_1$$

This optimization problem returns the optimized intermediate bridges between the drawn pairs. Subsequently, the parameterized drift $u_t^\theta$ is matched given the prescribed marginal path from the previous step, by minimizing the following variational gap

$$\min_\theta \int_0^1 \mathbb{E}_{p_t}[\|u_t^\star - u_t^\theta\|^2]dt \tag{5}$$

where $u_t^\star$ is the drift associated with the fixed intermediate path. Propagation of the SDE according to the matched parameterized drift $u_t^\theta : \mathbb{R}^d \times [0,1] \to \mathbb{R}$

$$dX_t = u_t^\theta dt + \sigma dW_t, \quad X_0 \sim \pi_0, \ X_1 \sim \pi_1 \tag{6}$$

induces the parameterized coupling $\pi_{0,1}^\theta$. In the next iteration of the BM algorithm, this updated coupling is used to generate improved pairs of $(x_0, x_1) \sim \pi_{0,1}^\theta$ employed to refine the marginal path.

## 3 METHODOLOGY

### 3.1 SEMI-SUPERVISED GUIDANCE

In the context of semi-supervised optimal transport, certain aligned couples of data points from the source and target distributions are employed to guide the matching of unpaired samples. These pairs are assumed by construction to be given by the optimal solution of the static SB. We denote with $\mathcal{K} = \{x_0^n, x_1^n\}_{n=1}^N$ the set with the aligned pairs, the Key-Point (KP) set, with $N$ being the number of pairs in the set. Usually, $N \ll M$, where $M$ is the total number of samples.

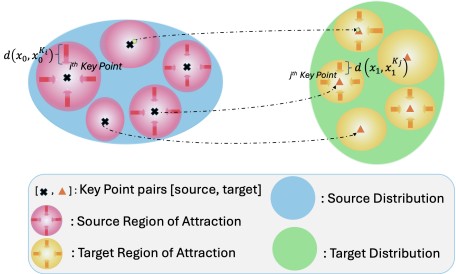

Figure 2: Source and Target Distributions with their KP set pairs

**Assumption 3.1.** All pairs from the KP set are considered to be solutions of the static Schrödinger Bridge.

Note that since the KP pairs are sampled from the solution of the static SB, this ensures their stochastic nature, making them well-suited for guiding the transport map. Our analysis begins with the static entropic OT problem in Eq. (3), to which an additional regularization term is included to capture the interaction among unpaired samples and aligned samples, thereby guiding the transport map. We consider a distance-preserving scheme. We start by sampling a pair of non-coupled samples $(X_0, X_1)$. Let us assume that the $i^{\text{th}}$ sample from the KP set $x_0^i \in \mathcal{K}$ is closest to $X_0$ in the source distribution. Then, the distance $d(X_0, x_0^i)$ is computed and we proceed to compute the distance $d(X_1, x_1^i)$. Therefore, for a fixed cou pair $(x_0^i, x_1^i) \in \mathcal{K}$, we define the Guidance function for the non-aligned pair $(X_0, X_1)$, and as

$$G(X_0, X_1) = (d(X_1, x_1^i) - d(X_0, x_0^i))^2 \tag{7}$$

Intuitively, this regularization term creates a region of attraction, as illustrated in Figure 2. This attraction around each KP sample defines clusters that guide the neighboring samples from the source distribution to the target distribution, by preserving the relative distance to the respective KP sample. Adjusting the cost function of the entropic O.T. formulation in 3, to incorporate guidance from the coupled pairs yields

$$\min_{\pi \in \Pi(\pi_0, \pi_1)} \int \left( ||X_0 - X_1||^2 + G(X_0, X_1) \right) d\pi(x_0, x_1) + \epsilon KL(\pi | \pi_0 \otimes \pi_1) \tag{8}$$

In this work, we start our analysis from the modified expression of Eq. (3) to exploit the flexibility of static OT formulation into adding the regularization term that is responsible for the guidance of the non-paired samples. However, our focal point is to derive an equivalent dynamic formulation, resembling Eq. (2), which provides a more thorough description of the transport process. From an algorithmic standpoint, our methodology is based on BM frameworks (see Section 2.2) in order to leverage their great efficiency and scalability in solving distribution matching problems.

### 3.2 FEEDBACK SCHRÖDINGER BRIDGE MATCHING

We propose **Feedback Schrodinger Brige Matching** (FBSM) a novel matching framework, which leverages the information of partially, optimally paired datasets to guide the matching of the non-aligned samples. Our analysis starts from recasting Eq. (8) in a dynamic formulation in Sec. 3.2.1. Subsequently, we adopt recent advances in dynamic SB frameworks, that propose a decomposition of the dynamic optimization problem into two components (Liu et al., 2024; Shi et al., 2023). Therefore, we separate the training phase into two stages: 1) the optimization of the intermediate path, conditioning on the endpoints $(x_0, x_1)$, and 2) the optimization of the parameterized drift, which results in refining the coupling.

### 3.2.1 DYNAMIC OBJECTIVE

The theorem below provides a general framework to recast the entropy regularized semi-supervised O.T. problem in Eq. (8) in a dynamic setting, regardless of the selection for the guidance function $G$.

**Theorem 3.2.** *Assume $X_0, X_1 \in \mathbb{R}^d$, with $X_0 \sim \pi_0$, and $X_1 \sim \pi_1$, and consider a stochastic random variable $X_t$ connecting $X_0$ and $X_1$, whose law is the continuous marginal probability path $p_t$ joining $\pi_0$, and $\pi_1$. Additionally consider the KP pairs $\{x_0^n, x_1^n\}_{n=1}^N$ with their fixed interpolating paths. Starting from the static semi-supervised guided entropic OT problem in Eq. (8), we derive the following relaxed dynamic formulation.*

$$\min_{p_t, u_t} \int_{\mathbb{R}^d \times [0,1]} \left( \frac{1}{2}\|u_t\|^2 + \mathbb{E}_{p_{0|t}}\left[ |u_t^\mathsf{T} \nabla G_{t,0}| + \frac{\sigma^2}{2}\Delta G_{t,0} \right] \right) p_t \mathrm{d}x \mathrm{d}t,$$

$$s.t. \ \frac{\partial p_t}{\partial t} = -\nabla \cdot (u_t p_t) + \frac{\sigma^2}{2}\Delta p_t, \ and \ p_0 = \pi_0, p_1 = \pi_1 \tag{9}$$

Notice that our dynamic formulation is a modified and generalized version of Eq. (2) to capture the guidance from the KP samples. For degenerate guidance term ($G \equiv 0$), one readily obtains the standard entropic analogue of the fluid dynamic formulation (Benamou & Brenier, 2000; Gentil et al., 2017) in Eq. (2) which also corresponds to the objective of SB objective (De Bortoli et al., 2021; Shi et al., 2023).

### 3.2.2 INTERMEDIATE PATH OPTIMIZATION

As discussed above, the initial step of our methodology entails the optimization of the intermediate path. Let the marginal be expressed as a mixture of conditional probability paths $p_t = \int p_{t|0,1} \pi_{0,1} \mathrm{d}x_0 \mathrm{d}x_1$. More explicitly, we sample non-coupled pairs $(x_0, x_1)$ and fix the coupling at the boundaries $\pi_{0,1}$. This results in relaxing the distributional constraints in Eq. (9) to just two endpoints. The optimization of the intermediate path yields the optimal marginal conditioned on the two endpoints $p_{t|0,1}$ of the uncoupled samples, which crucially preserves their relative distance to the KPs throughout the entire trajectory.

**Proposition 3.3.** *Let the continuous marginal path $p_t$ satisfy the following decomposition $p_t = \int p_{t|0,1} \pi_{0,1} \mathrm{d}x_0 \mathrm{d}x_1$. For optimizing with respect to the intermediate path, the coupling $\pi_{0,1}$ is frozen which results in Eq. (9) being recasted as*

$$\min_{p_{t|0,1}} \int_{\mathbb{R}^d \times [0,1]} \left( \frac{1}{2}\|u_{t|0,1}\|^2 + |u_{t|0,1}^\mathsf{T} \nabla G(X_t, x_0)| + \frac{\sigma^2}{2}\Delta G(X_t, x_0) \right) p_{t|0,1} \mathrm{d}x \mathrm{d}t,$$

$$s.t. \ \frac{\partial p_{t|0,1}}{\partial t} = -\nabla \cdot (p_{t|0,1} u_t) + \frac{1}{2}\sigma^2 \Delta p_{t|0,1}, \quad X_0 = x_0, \quad X_1 = x_1 \tag{10}$$

Importantly, the conditioning on the unpaired sample $x_0$ acts as an anchor. Our guidance function $G$ groups the unpaired data into clusters around the KPs in the source distribution, and penalize the deviation of the unpaired samples from the trajectory that their assigned keypoint sample follows, preserving structural and distance information in the dynamic transport map. This forces unpaired samples to preserve their relative distance from the closest KP sample in the source distribution at each time-step.

*Remark* 3.4. Notice that in the extreme case in which the entire dataset is comprised of only pre-aligned couples, $G$ would be trivial, and Eq. (10) would retrieve the diffusion bridges between optimal pairs, similarly to $I^2SB$ (Liu et al., 2023). Conversely, in the other extreme case, where the KP set is the empty set, our algorithm retrieves DSBM.

**Feedback** Interestingly, in stochastic optimal control literature ((Theodorou et al., 2010)) it has been reported that the inner product of the drift with the state cost acts as state feedback. Hence, the term $|\nabla G_{t|0}^\mathsf{T} u_{t|0,1}|$ is interpreted as state-feedback that steers the non-coupled samples, by projecting the conditioned drift on the gradient of $G_{t|0}$ Figure 3 illustrates the gradient field of the guidance function, and how it pushes the optimal trajectory towards the path of the aligned data. Note that the trajectories of the KP samples along with their coupling are assumed to be given and fixed for each KP pair. Furthermore, the absolute value implies that the angle of projection is always acute, which greatly improves stability during training.

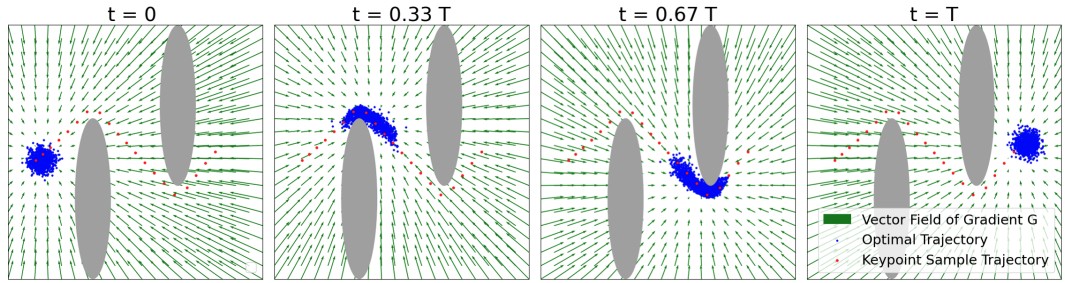

Figure 3: Gradient Field of $\nabla G$ in the S-tunnel

**Intemediate Path** Recall that we have relaxed the distributional constraints in Eq. (9) to just two endpoints. We solve Eq. (10) for each pair $(x_0, x_1) \sim p_{0,1}^\theta$ and later marginalize to construct $p_t$. The conditioned probability path is approximated in a simulation-free manner as a Gaussian path $p_{t|0,1} \approx \mathcal{N}(I_{t|0,1}, \sigma_t \mathbf{I}_d)$, where $I_{t|0,1} \equiv I(t, x_0, x_1)$ is the interpolant function between the pinned endpoints $I_0 = x_0$, $I_1 = x_1$, and $\sigma$ is the standard deviation with $\sigma_0 = \sigma_1 = 0$. This enables us to compute a closed-form expression for the conditional drift of the uncoupled pairs (Särkkä & Solin, 2019; Albergo et al., 2023).

$$u_{t|0,1} = \partial_t I_{t|0,1} + \Big(\frac{\partial_t \sigma_t}{\sigma_t} - \frac{\nu^2}{2\sigma_t^2}\Big)(X_t - I_t) \tag{11}$$

where $\nu$ is by definition of $\sigma_t = \nu\sqrt{t(1-t)}$. More details are left for Appendix B.1.

### 3.2.3 COUPLING OPTIMIZATION

Subsequently, we will match the parameterized drift $u_t^\theta$, given the prescribed path $p_{t|0,1}$ of the uncoupled samples from the previous step, which will further improve the intermediate path in the next iteration. The more general form of the Lagrangian in Eq. (10) prompts us to express the variational gap through the Bregman divergence, following recent advancements in bridging optimality gaps for general Lagrangian costs Neklyudov et al. (2023). However, before deriving the expression of the variational gap, we need the convex conjugate of the Lagrangian in Eq. (10).

**Proposition 3.5.** *The convex conjugate of the Lagrangian in Eq. (10) is defined as the Hamiltonian $H(x_t, a_t, t) = \sup_{u_{t|0,1}} \langle u_{t|0,1}, a_t \rangle - L(x_t, u_{t|0,1}, t)$. The optimization with respect to the drift yields $u_{t|0,1} = a_t - g(a_t^\intercal \nabla G_{t|0}) \nabla G_{t|0}$, where $\nabla G_{t|0} \equiv G(X_t, x_0)$, and $g(\cdot) : \mathbb{R} \to \mathbb{R}$ is given by*

$$g(a_t^\intercal \nabla G_{t|0}) = \begin{cases} \frac{a_t^\intercal \nabla G_{t|0}}{\|\nabla G_{t|0}\|^2} & \text{if} \quad |a_t^\intercal \nabla G_{t|0}| \leq \|\nabla G_{t|0}\|^2 \\ sgn(a_t^\intercal \nabla G_{t|0}) & \text{else} \end{cases}$$

*Finally, we obtain the Hamiltonian associated with the Lagrangian in Eq. (10)*

$$H(x_t, a_t, t) = \frac{1}{2}\|a_t - g(a_t^\intercal \nabla G_{t|0}) \cdot \nabla G_{t|0}\|^2 - \frac{\sigma^2}{2}\Delta G(X_t, x_0) \tag{12}$$

At this point, we introduce our Entropic Lagrangian Bridge Matching objective through the Bregman divergence, using the Hamiltonian in Eq. (12).

**Proposition 3.6.** *Consider the optimized conditional drift $u_{t|0,1}^\star$ following the optimization in Eq. (10). The parameterized drift $u_t^\theta$ is matched to $u_{t|0,1}^\star$ given the prescribed marginal path by minimizing the variational gap expressed through the Bregman divergence: $D_{L,H} = L(X_t, u_t, t) + H(X_t, a_t, t) - \langle a_t, u_t \rangle$*

$$\min_\theta \int_0^1 \mathbb{E}_{p_{0,1}} \mathbb{E}_{p_{t|0,1}} \|a_t^\theta - u_{t|0,1}^\star - g(a_t^\intercal \nabla G_{t|0}) \cdot \nabla G_{t|0}\|^2 \mathrm{d}t \tag{13}$$

*From Proposition 3.5, the parameterized drift can be expressed as $u_t^\theta = a_t^\theta - g(a_t^\intercal \nabla G_{t|0}) \cdot \nabla G_{t|0}$. Therefore, the matching objective of Eq. (13) can be rewritten as the objective in Eq. (5)*

$$\min_\theta \int_0^1 \mathbb{E}_{p_t} \|u_{t|0,1}^\star - u_t^\theta\|^2 \mathrm{d}t \tag{14}$$

---

**Algorithm 1** Feedback Schrödinger Bridge Matching (**FSBM**)

---

1: **Input**: Boundary distributions $\pi_0$, and $\pi_1$ and the pairs from the key-point (KP) set $\{x_0^n, x_1^n\}_{n=1}^N$, and their fixed trajectories
2: Initialize $u_t^\theta$, and $\pi_{0,1}$ as independent coupling $\pi_0 \otimes \pi_1$.
3: **repeat**
4:     Sample $x_0, x_1, X_{t_k}$ by forward propagating using $u_t^\theta$, on time steps $0 < t_1 < \cdots < t_K < 1$
5:     Find closest aligned sample $x_0^i$ to $x_0$, and calculate $d(x_0, x_0^i)$
6:     Using the same $i^{\text{th}}$ data point from the KP set, calculate $d(X_{t_k}, x_{t_k}^i)$
7:     $G(X_{t_k}, x_0)$ from Eq. (7)
8:     Fix the $x_0, x_1$, and obtain $p_{t|0,1}^\star$ using Eq. (10)
9:     Calculate $u_{t|0,1}$ from Eq. (11)
10:    Update $u_t^\theta$, from Eq. (14)
11: **until** converges

---

### 3.3 TRAINING SCHEME

A summary of our training procedure is presented in Alg. 1. Our algorithm finds a solution $(p_t^\star, u_t^\star)$ to Eq. (9) by iteratively optimizing Eq. (10), and Eq. (14). First, we draw pairs $(x_0, x_1)$, and fix the parameterized coupling $\pi_{0,1}^\theta$. Subsequently, to compute the guidance function, let the $i^{\text{th}}$ data point from the KPs set $x_0^i \in \mathcal{K}$ be closest to $x_0$ in the source distribution, the distance $d(x_0, x_0^i)$ between the two samples is evaluated. Next, for every sampled time-step $t_k$, we compute the distance $d(X_{t_k}, X_{t_k}^i)$ between $X_{t_k}$, and $X_{t_k}^i$, and evaluate $G(X_{t_k}, x_0)$. Then, optimizing Eq. (10), between the endpoints $(x_0, x_1)$ from the first step, yields the optimal conditional path $p_{t|0,1}^\star$, whose corresponding drift $u_{t|0,1}^\star$ is obtained from Eq. (11). This drift ensures that the uncoupled data follow the trajectory of the KPs. Lastly, we fix the marginal path and optimize the coupling, by matching the parameterized drift $u_t^\theta$ to the optimized drift of the uncoupled samples $u_{t|0,1}^\star$, using Eq. (14). The optimized parameterized drift $u_t^\theta$ induces an improved coupling $\pi_{0,1}^\theta$ by propagating the dynamics through the SDE in Eq. (6).

## 4 EXPERIMENTS

We demonstrate the efficacy of our FSBM in a variety of distribution matching tasks, such as Crowd Navigation, Opinion Depolarization, and Unpaired Image Translation, compared against other state-of-the-art distribution matching frameworks such as GSBM (Liu et al., 2024), DSBM (Shi et al., 2023), and Light and Optimal Schrodinger Bridge Matching (LOSBM; Gushchin et al. (2024)).

### 4.1 CROWD NAVIGATION

We first showcase the efficacy of our FSBM in solving complex crowd navigation tasks and its superior capability to generalize under a variety of initial conditions, listed as: **i) Vanilla**: initial distribution is the same as the one the model was trained on, **ii) Perturbed Mean**: shift the mean of Vanilla, **iii) Perturbed STD**: shift the standard deviation (STD) of Vanilla, **iv) Uniform Distribution**: the initial distribution is defined as a uniform distribution. We compare our FSBM with the GSBM framework (Liu et al., 2024). The aligned pairs used to guide the transport map were less than 4% of the uncoupled pairs for both tasks. More details are left for Appendix C.1.

Table 1 shows that FSBM generated more accurately the targeted distribution in both tasks. Additionally, Figure 4 depicts our FSBM consistently producing better trajectories, successfully avoiding the obstacles under all initial conditions, and generating final distributions $p_1^\theta$, that more closely match the target distribution $\pi_1$, also shown by the smaller values of the Wasserstein distance $\mathcal{W}_2(p_1^\theta, \pi_1)$. In contrast, under GSBM, several particles were observed to diverge completely in the S-tunnel task under shifted initial conditions, causing the increase of the $\mathcal{W}_2$ distance. Additionally, notice that the final distributions generated by GSBM (see navy blue particles in the upper row of Figure 4), even for the samples that did not diverge, were substantially more inaccurate when compared to our FSBM. Lastly, Fig. 5 illustrates that our FSBM required significantly less wall-clock time for training. This is attributed to the faster convergence of FSBM with regards to the number of epochs, but also due to GSBM requiring the computation of the entropy and congestion cost, which according to our experiments has been shown to increase training duration significantly.

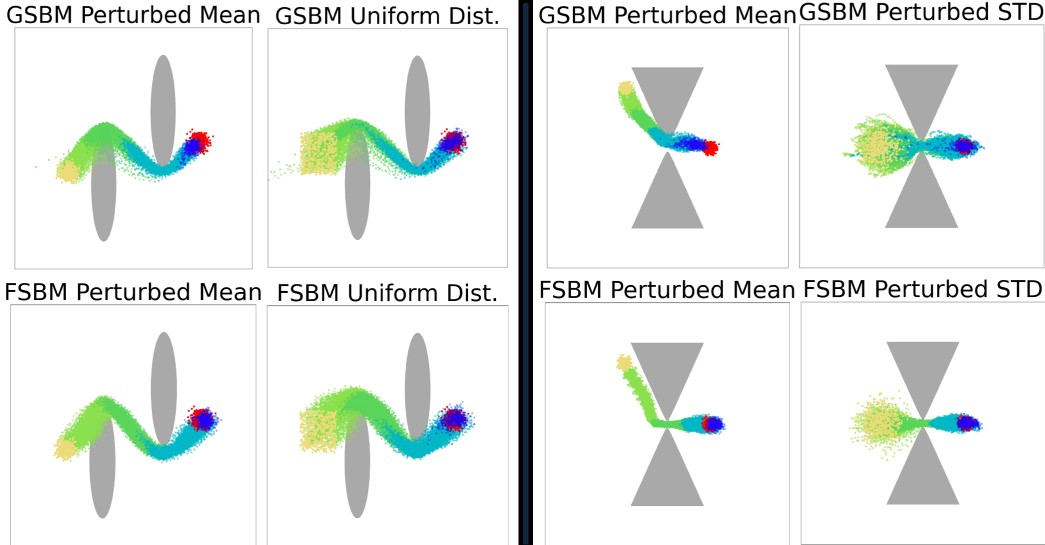

Figure 4: Crowd Navigation comparison between GSBM and FSBM in the S-tunnel, and V-neck under perturbed mean and uniform initial distribution, and in the V-neck under perturbed mean and perturbed STD. The color of particles means: **i) Yellow:** initial conditions, **ii) Green and Cyan:** intermediate trajectory, **iii) Red:** target distribution, **iv) Navy Blue:** generated distribution

Table 1: $\mathcal{W}_2$ distance between generated and ground truth distributions for GSBM and FSBM in (Lower is better)

|  | S-tunnel | | V-neck | |
|---|---|---|---|---|
|  | GSBM | FSBM | GSBM | FSBM |
| Vanilla | 0.02 | 0.02 | 0.01 | 0.01 |
| Perturbed Mean | 8.87e+5 | **0.67** | 0.68 | **0.34** |
| Perturbed STD | 2.43e+10 | **0.34** | 3.03 | **0.11** |
| Uniform Distribution | 6.65e+9 | **0.49** | 0.10 | **0.10** |

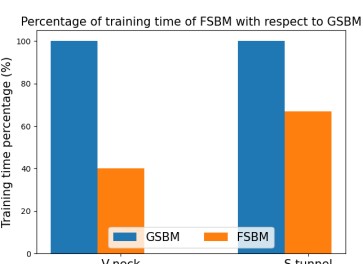

Figure 5: Training time percentage comparison between GSBM and FSBM in Crowd Navigation tasks

## 4.2 OPINION DYNAMICS

Subsequently, we consider high-dimensional opinion depolarization, where each particle possesses a high-dimensional opinion $X_t \in \mathbb{R}^{100}$ that evolves through interaction with the other agents under polarizing dynamics which tend to segregate them into two groups of diametrically opposing opinions (see Figure 6: first row, second column). More information about the polarization base drift are left in Appendix C.2. To mitigate the segregation, we apply our FSBM method employing the KPs samples to guide the rest of the population to a more uniform opinion, closer to the target distribution (see Figure 6: second row third column). We compare against DeepGSB and GSBM (Liu et al., 2022a; 2024), which employ state cost to mitigate polarizing dynamics. Lastly, it is worth noting that the size of the KP set utilized for the opinion depolarization was approximately 2%, of the entire population.

The results shown demonstrate that the utilization of certain opinion leaders, that guide the overall opinion of the population to depolarization renders our FSBM a highly efficient model to mitigate opinion segregation. The metrics used to evaluate the proximity of the generated and target distributions (e.g. $p_1^\theta$, $\pi_1$ respectively) are the $\mathcal{W}_2(p_1^\theta, \pi_1)$ distance and the KL divergence. Observation of Table 2 and Figure 6 indicates that our FSBM retrieves a solution that is slightly closer to the targeted distribution compared to GSBM, albeit needing half of the training time.

Table 2: Comparison of $\mathcal{W}_2(p_1^\theta, \pi_1)$ distance and KL divergence between target and generated distribution in High Dimensional ($X \in \mathbb{R}^{100}$) Opinion Dynamics among Deep GSB, GSBM and FSBM (ours) (Lower is better in all metrics)

| Optimizer | Time (hours:min) | W2 Distance | KL divergence |
|-----------|------------------|-------------|---------------|
| DeepGSB | 4:37 | 58.16 | 0.096 |
| GSBM | 5:29 | 43.41 | **0.034** |
| FSBM | **2:43** | **42.91** | 0.034 |

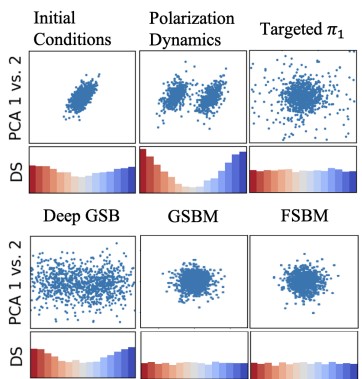

Figure 6: Opinion depolarization from our FSBM and GSBM, and DeepGSB

## 4.3 IMAGE TRANSLATION

Finally, we present the results of our FSBM in two types of image translations: **i)** Gender translation, **ii)** Age translation. We follow the setup of (Korotin et al., 2023) with the pre-trained ALAE autoencoder (Pidhorskyi et al., 2020) on the 1024 ×1024 FFHQ dataset (Karras et al., 2019) to perform the translation in the latent space of dimensions $512 \times 1$, enabling more efficient training and sampling. Notably, the number of aligned images was $4\%$ of the total dataset for the gender translation and $8\%$ for the age translation. Our method is compared with other state-of-the-art matching frameworks, such as DSBM (Shi et al., 2023) and Light and Optimal SBM (LOSBM)(Gushchin et al., 2024). Additional details are left for Appendix C.3.

Table 3: FID values in 4 translation tasks for DSBM, LOSBM, and our FSBM (Lower is better)

| Optimizer | Man to Woman | Woman to Man | Old to Young | Young to Old |
|-----------|--------------|--------------|--------------|--------------|
| DSBM | 15.90 | 17.02 | 16.83 | 17.76 |
| LOSBM | 18.89 | 17.24 | 17.89 | 19.32 |
| FSBM | **14.78** | **16.12** | **15.52** | **17.14** |

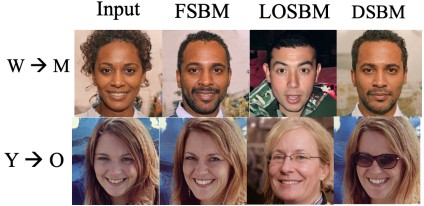

Figure 7: Translation comparison between our FSBM, LOSBM and DSBM

Table 3 shows that our FSBM consistently achieves lower FID value, than the other methods. Additionally, it is demonstrated that our FSBM recovers better couplings through the qualitative comparison in Figure 7 and quantitatively through the LPIPS values in Table 5. We can see that DSBM and FSBM return very close LPIPS values, along with very similar images, which suggests that the two methods converge to close solutions. Conversely, the translations of LOSBM in the validation set were losing some of the features of the input image. Figure 8 presents additional examples of the couplings obtained from our FSBM. Our FSBM demonstrates highly accurate translations preserving key features identical, such as smile, accessories, angle of the face, and background color. Notably, Table 4 demonstrates that FSBM required *only an additional 180 MB of VRAM* compared to DSBM, to manage the aligned latent vectors, which is tractable for most modern GPUS, while training DSBM and FSBM for the same number of epochs requires *virtually identical training duration*.

## 4.4 LIMITATIONS AND DISCUSSIONS

Our FSBM has demonstrated remarkable results in leveraging partially aligned datasets for distribution matching. However, our framworks relies heavily on the availability and the quality of the available KP pairs. The manner in which we sample from solution of the static SB does not ensure that all modes are equally and satisfactorily represented. This implies that in cases where a minimal number

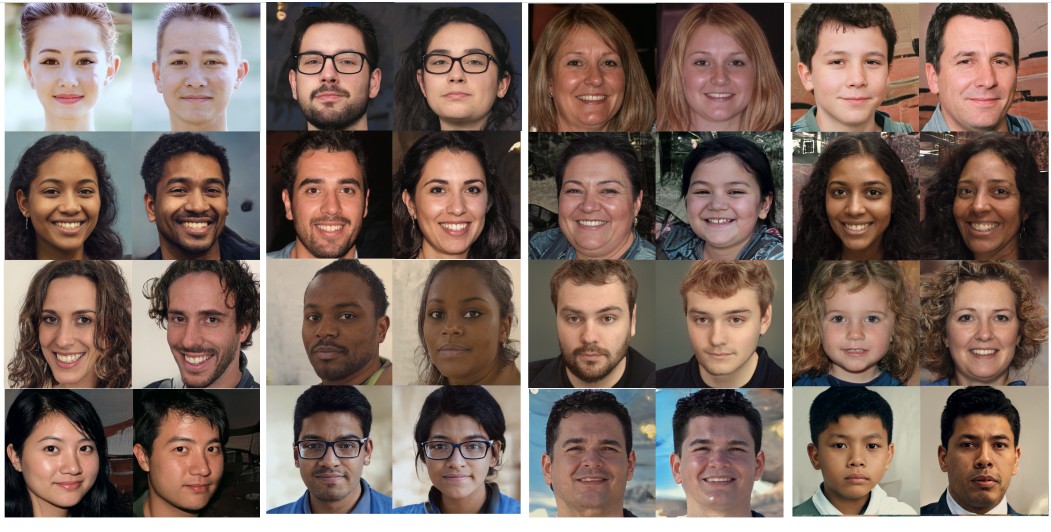

Woman to Man    Man to Woman    Old to Young    Young to Old

Figure 8: Image Translations using our FSBM

Table 4: Epoch duration and memory usage (GB) of different stages in Alg. 1, measured on the age translation task

|  | Epoch Time (hours:min) | Memory (GB) |
|---|---|---|
| *Guidance (lines 5-8)* | *00:01* | *0.18* |
| Matching (lines 9-11) | 02:56 | 1.51 |

Table 5: LPIPS values in 4 translation tasks for DSBM, LOSBM, and our FSBM (Lower is better)

| Optimizer | Man to Woman | Woman to Man | Old to Young | Young to Old |
|---|---|---|---|---|
| DSBM | 0.47 | 0.48 | 0.53 | 0.49 |
| LOSBM | 0.70 | 0.69 | 0.71 | 0.70 |
| FSBM | **0.44** | **0.43** | **0.48** | **0.47** |

of these pairs are available, the regions of attraction formed around these may not sufficiently cover the diversity of the source distribution. This could lead to wrongful guidance, suboptimal matching and poor generalizabilty.

In future work, we have identified several directions which could improve our work even further. For instance, we aim to devise a more intricate mechanism to sample from the solution of the static SB, while guaranteeing that all modes in highly diverse datasets are covered and represented through the sampled KPs. Additionally, we aim to further explore the capabilities of the guidance function and devise more task-specific function which furhter improve perfomance generalizability by optimizing the information offered by the KP samples. Additionally, we intend to improve the way we impose guidance, by allowing the aligned pairs to preserve the relative distance to k-nearest neighboring KPs, instead of simply the nearest one.

## 5 CONCLUSION

In this work, we developed FSBM, a novel matching algorithm capable of using the information from pre-aligned data pairs, to guide the transport mapping of samples from the source to the target distribution. We demonstrated our algorithm's enhanced generalizability and reduced training over prior methods through extensive experimentation. In future work, we wish to delve deeper into the effect of the KP samples on the matching procedure, along with using more intricate and task-specific guidance functions for further improvements. In summary, our work paves new ways for training diffusion models with partially aligned data.

ACKNOWLEDGEMENTS

The authors would like to thank Tianrong Chen and Kevin Rojas for the useful and fruitful discussions.

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

## A APPENDIX

### A.1 PROOF OF THEOREM 3.2

*Assume $X_0, X_1 \in \mathbb{R}^d$, with $X_0 \sim \pi_0$, and $X_1 \sim \pi_1$, and consider a stochastic random variable $X_t$ connecting $X_0$ and $X_1$, whose law is the continuous marginal probability path $p_t$ joining $\pi_0$, and $\pi_1$. Starting from the static semi-supervised guided entropic OT problem in Eq. (8), we modify it appropriately and obtain the following relaxed dynamic formulation.*

$$\min_{p_t, u_t} \int_{\mathbb{R}^d \times [0,1]} \left( \frac{1}{2} \|u_t\|^2 + \mathbb{E}_{p_{0|t}} \left[ |u_t^\mathsf{T} G_{t,0}| + \frac{\sigma^2}{2} \Delta G_{t,0} \right] \right) p_t \mathrm{d}x \mathrm{d}t,$$

$$\text{s.t. } \frac{\partial p_t}{\partial t} = -\nabla \cdot (u_t p_t) + \frac{\sigma^2}{2} \Delta p_t, \text{ and } p_0 = \pi_0, p_1 = \pi_1 \tag{15}$$

Before the proof, we provide a sketch of the proof overviewing the key steps towards the derivation of our dynamic formulation.

*Sketch of Proof.* Let $\pi_0, \pi_1$ be the two given boundary laws, recall the static formulation: $\min_\pi \int \|X_0 - X_1\|^2 + G(X_0, X_1) + \epsilon KL(\pi_{0,1} | \pi_0 \otimes \pi_1)$. The first step of our analysis is a re-arrangement of the terms of the static semi-superivsed EOT problem in Eq. (8). This rearrangement allows us to rewrite the modified EOT as a regularized static Schrodinger Bridge problem.

$$\min_\pi KL(\pi_{0,1} | \pi_{0,1}^\mathbb{Q}) + \int G(X_0, X_1) \pi_{0,1} dx_0 dx_1$$

Starting from the EOT problem, from $t_0 = 0$ and $t_N = 1$, we can consider any intermediate point $t_1$ and find the optimal paths $p_1^\star, p_2^\star$ between the endpoints in the two newly formed subproblems (i.e., from $t_0$ to $t_1$, and $t_1$ to $t_N$). The union of the optimal subpaths forms the optimal policy of the initial problem, and the converse is also true Villani et al. (2009). This can be extended for any number $N$ of intermediate points. As $N \to \infty$, this leads us to to consider 'local' versions of optimal transport problems between infinitesimally close distributions. This implies that the local path $p_{t_i, t_{i+1}}$ converges to the continuous marginal $p_t$, for arbitrarily large number of steps in $[0, 1]$ Villani et al. (2009). Therefore, in the next step, we express the guidance constraint EOT problem as a sequence of local EOT problems.

$$\min_{p_t} \min_{X_t} \sum_{i=0}^{N-1} KL(p_{t_{i \to i+1}} | q_{i \to i+1}) + \mathbb{E}_{p_t} \sum_{i=0}^{N-1} G(X_{t_i}, X_{t_{i+1}})$$

Subsequently, we rewrite the $G : \mathbb{R}^d \times \mathbb{R}^d \to \mathbb{R}$ function between $X_{t_i}$ and $X_{t_{i+1}}$, and as a sum of telescopic series with constant reference point $X_0$ from the source distribution. This induces the difference $G(X_{t_{i+1}}, X_0) - G(X_{t_i}, X_0)$. Although, this term is positive at the optimal path, in order to ensure positivity of this term throughout the optimization process, we upper bound it with its absolute value.

$$\min_{p_t, X_t} \sum_{i=0}^{N-1} \left| \mathbb{E}_{p_{t_{i,i+1,0}}} G(X_{t_i}, X_0) - G(X_{t_{i+1}}, X_0) \right| + \sum_{i=0}^{N-1} KL(p_{t_{i \to i+1}} | q_{i \to i+1})$$

Finally, we apply the Girsanov theorem in the KL divergence and Ito's Lemma in the difference $G(X_{t_{i+1}}, X_0) - G(X_{t_i}, X_0)$ introduced in step 3, where the stochastic terms are dropped assuming zero mean Wiener noise.

Assuming now that the number of intermediate steps $N \to \infty$, this yields our continuous dynamic formulation

$$\min_{p_{t|0,1}} \int \mathbb{E}_{p_t} \left[ \frac{1}{2} \|u_{t|0,1}\|^2 + \left[ |u_{t|0,1}^\mathsf{T} \nabla G_{t|0} + \frac{\sigma^2}{2} \Delta G_{t|0}| \right] \right], \text{ s.t. } \frac{\partial p_{t|0,1}}{\partial t} = -\nabla \cdot (u_{t|0,1} p_{t|0,1}) + \frac{\sigma^2}{2} \Delta p_{t|0,1}$$

*Proof.* First, let $c(X_0, X_1)$ be the cost from the static optimal transport expression, and let $\pi_0, \pi_1$ be the two given boundary laws. Recall the static formulation: $\min_\pi \int \|X_0 - X_1\|^2 + G(X_0, X_1) + \epsilon KL(\pi_{0,1}|\pi_0 \otimes \pi_1)$. Our objective is to derive a dynamic formulation for the static EOT problem with the cost function $c(X_0, X_1) = \|X_0 - X_1\|^2 + G(X_0, X_1)$. We can rewrite the divergence $KL(\pi_{0,1}|\pi_0 \otimes \pi_1)$, and the quadratic cost $\|X_0 - X_1\|^2$, as $KL(\pi_{0,1}|\pi_{0,1}^\mathbb{Q})$ assuming a gaussian prior measure $\mathbb{Q}$ (Nutz, 2021). More specifically, it holds that $\log \pi_{0,1}^\mathbb{Q} \propto \|X_0 - X_1\|^2/\epsilon$, hence $\pi^\mathbb{Q} \propto e^{\|X_0 - X_1\|^2/\epsilon}$. We rewrite the objective in Eq. (16), as $KL(\pi_{0,1}|\pi_{0,1}^\mathbb{Q}) + \int G(x_0, x_1)\pi_{0,1}dx_0dx_1$. Therefore, the static objective can be rewritten as

$$\min_\pi KL(\pi_{0,1}|\pi_{0,1}^\mathbb{Q}) + \int G(X_0, X_1)\pi_{0,1}dx_0dx_1 \tag{16}$$

In this work, our focal point is the derivation of a dynamic expression for the semi-supervised regularized cost function in 8. Through the prism of entropic interpolation Gentil et al. (2017), the solution of dynamical OT formulations correspond to solving 'local' versions of optimal transport problems between infinitesimally close distributions Villani et al. (2009). Let a stochastic random variable path $(X_t)_{0 \le t \le 1}$ joining $X_0 \sim \pi_0$ to $X_1 \sim \pi_1$, namely $X_t$ is an interpolation from $X_0$ to $X_1$. This random variable is assumed to be the solution of the following SDE

$$dX_t = u_t dt + \sigma dW_t \tag{17}$$

Equivalent we can consider a marginal density induced by Eq.17 $(p_t)_{0 \le t \le 1}$ which joins $\pi_0$, and $\pi_1$ Villani et al. (2009) admits the Fokker Plank Equation (FPE):

$$\frac{\partial p_t}{\partial t} = -\nabla \cdot (u_t p_t) + \frac{1}{2}\sigma^2 \Delta p_t \tag{18}$$

Now, application of [Villani et al. (2009), Theorem 7.21] suggests that if we consider $p_t$ be the solution to the infimum above, namely a minimizing curve. The endpoints of the random curve $X_t$, whose law admits an optimal transference plan, are an optimal coupling $(X_{t_i}, X_{t_{i+1}})$. Then from the definition of the optimal cost and the minimizing property of the $X_t$

$$\min_{p_t} \min_{X_t} \sum_{i=0}^{N-1} KL(p_{t_{i \to i+1}}|q_{i \to i+1}) + \mathbb{E}_{p_t} \sum_{i=0}^{N-1} G(X_{t_i}, X_{t_{i+1}}) \tag{19}$$

for with $p_0 = \pi_0$, and $p_N = \pi_N$. It generally holds $\mathbb{E}_{p_t} c(x_i, x_{i+1}) \ge \mathbb{E}_{p_t} c(x_0, x_{i+1}) - \mathbb{E}_{p_t} c(x_0, x_i)$, with the equality being true for the minimizing path. Therefore, we can say that

$$\min_{p_t} \min_{X_t} \mathbb{E}_{p_t} c(X_i, X_{i+1}) = \min_{p_t} \min_{X_t} \mathbb{E}_{p_t} c(X_0, X_{i+1}) - \mathbb{E}_{p_t} c(X_0, X_i) \tag{20}$$

This yields the modified version of the dynamic analogue

$$\min_{p_t} \min_{X_t} \sum_{i=0}^{N-1} KL(p_{t_{0,i+1}}|q_{0,i+1}) - KL(p_{t_{0,i}}|q_{0,i})$$
$$+ \sum_{i=0}^{N-1} \mathbb{E}_{p_{t_{i+1},0}}[G(X_{t_{i+1}}, X_0)] - \mathbb{E}_{p_{t_i},0}[G(X_{t_i}, X_0)] \tag{21}$$

However, note that $KL(p_{t_{0,i+1}}|q_{0,i+1}) - KL(p_{t_{0,i}}|q_{0,i}) \le KL(p_{t_{i \to i+1}}|q_{i \to i+1})$. Subsequently, note for any coupling between $p_{t_i}$, and $p_{t_{i+1}}$, it holds that

$$\mathbb{E}_{p_{t_{i+1},0}}[c(X_{t_{i+1}}, X_0)] - \mathbb{E}_{p_{t_i},0}[c(X_{t_i}, X_0)] \le \mathbb{E}_{p_{t_{i \to i+1},0}}[c(X_{t_i}, X_0) - c(X_{t_{i+1}}, X_0)]$$

Finally, in order to ensure positivity of the difference $G(X_{t_i}, X_0) - G(X_{t_{i+1}}, X_0)$, we upper bound with its absolute value, and finally obtain the relaxed dynamic objective, which admits the following formulation

$$\min_{p_t, X_t} \sum_{i=0}^{N-1} \left| \mathbb{E}_{p_{t_{i,i+1},0}} G(X_{t_i}, X_0) - G(X_{t_{i+1}}, X_0) \right| + \sum_{i=0}^{N-1} KL(p_{t_{i \to i+1}}|q_{i \to i+1}) \tag{22}$$

For the first term, assuming the Markovian property of the base measure, we apply the Girsanov theorem in the infinitesimally small time-step $i \to i+1$:

$$KL(p_{i,i+1}|q_{i,i+1}) = \int_{t_i}^{t_{i+1}} \|u_{t_i}\|^2 dt = \|u_{t_i}\|^2 \Delta t_i \tag{23}$$

For the second term, we have

$$\left| \mathbb{E}_{p_{t_{i \to i+1,0}}}[G(X_{t_{i+1}}, X_0) - G(X_{t_i}, X_0)] \right|$$

For discretized random variable $X_{t_i}$ obeying the following discretized SDE: $X_{t_{i+1}} = X_{t_i} + u_{t_i}\Delta t_i + \sigma_{t_i}\Delta W_{t_i}$, application of the discrete version of the Ito's Lemma yields

$$\sum_{i=0}^{N-1} \left| \mathbb{E}_{p_{t_{i \to i+1,0}}}[(\frac{\partial G_{t_i}}{\partial t} + u_t^\intercal \nabla G_{t_i} + \frac{1}{2}\sigma_{t_i}^2 \text{Tr}(\nabla^2 G_{t_i}))\Delta t_i + (\sigma_{t_i}\nabla G_{t_i})\Delta W_{t_i}] \right| \tag{24}$$

It holds that $\frac{\partial G_{t_i}}{\partial t_i} = 0$, since $G$ does not explicitly depend on $t$, and also assume that the Wiener process increment has zero expectation, hence the equation above is rewritten as

$$\sum_{i=0}^{N-1} \left| \mathbb{E}[(u_t^\intercal \nabla G_{t_i} + \frac{1}{2}\sigma_{t_i}^2 \text{Tr}(\nabla^2 G_{t_i}))\Delta t_i] \right| \tag{25}$$

Now, we make use of the fact that $|\mathbb{E}[f(x)]| \le \mathbb{E}[|f(x)|]$, to take the expectation out of the absolute value.

$$\sum_{i=0}^{N-1} \left| \mathbb{E}\left[u_t^\intercal \nabla G + \frac{\sigma^2}{2}\text{Tr}(\nabla^2 G_{t_i})))\Delta t\right] \right| \le \sum_{i=0}^{N-1} \mathbb{E}\left[|(u_t^\intercal \nabla G + \frac{\sigma^2}{2}\text{Tr}(\nabla^2 G_{t_i})))|\Delta t\right]$$
$$\le \sum_{i=0}^{N-1} \mathbb{E}\left[(|u_t^\intercal \nabla G| + |\frac{\sigma^2}{2}\text{Tr}(\nabla^2 G_{t_i}))\Delta t\right] \tag{26}$$

Note that $G$ is convex, i.e., $\frac{\sigma^2}{2}\text{Tr}(\nabla^2 G) \ge 0$, thus we conclude that the expression in Eq. (33) is upper bound by

$$\mathbb{E}_{p_{t_{i \to i+1,0}}} \sum_{i=0}^{N-1} G(X_{t_i}, X_{t_{i+1}}) \le \sum_{i=0}^{N-1} \mathbb{E}_{p_{t_{i \to i+1,0}}}\left[(|u_t^\intercal \nabla G| + \frac{\sigma^2}{2}\text{Tr}(\nabla^2 G_{t_i}))\Delta t\right] \tag{27}$$

Finally, let us substitute back in the expression above the function $G$, and consider the continuous analogue of Eq. (23) and 27

$$\min_{p_t} \min_{u_t} \int_0^1 \mathbb{E}_{p_t}[\|u_t\|^2] + \mathbb{E}_{p_{t,0}}\left[(|u_t^\intercal \nabla G_{t,0}| + \frac{\sigma^2}{2}\Delta G_{t,0})\right]dt \tag{28}$$

where $G_{t,0} = G(X_t, X_0)$. Finally, we conclude the relaxed dynamic formulation is given by

$$\min_{p_t} \min_{u_t} \int_0^1 \mathbb{E}_{p_t}\left[\frac{1}{2}\|u_t\|^2 + \mathbb{E}_{p_{0|t}}[|u_t^\intercal \nabla G_{t,0}| + \frac{\sigma^2}{2}\Delta G_{t,0}]\right]dt \tag{29}$$

subjected to the FPE: $\frac{\partial p_t}{\partial t} = -\nabla \cdot (u_t p_t) + \frac{1}{2}\sigma^2 \Delta p_t$, and under the boundary constraints: $p_0 = \pi_0$, and $p_1 = \pi_1$. □

## A.2 PROOF OF PROPOSITION 3.3

*Let the continuous marginal path $p_t$ satisfy the following decomposition $p_t = \int p_{t|0,1}\pi_{0,1}dx_0 dx_1$. For optimization with respect to the intermediate path, we freeze the joint distribution at the boundaries $\pi_{0,1}$, which results in Eq. (9) being recasted as*

$$\min_{p_{t|0,1}} \int_0^1 \int_{\mathbb{R}^d} (\frac{1}{2}\|u_{t|0,1}\|^2 + |u_{t|0,1}^\intercal \nabla G(X_t, x_0)| + \frac{\sigma^2}{2}\Delta G(X_t, x_0) p_{t|0,1}dxdt,$$
$$\text{s.t. } \frac{\partial p_{t|0,1}}{\partial t} = -\nabla \cdot (p_{t|0,1}u_t) + \frac{1}{2}\sigma^2 \Delta p_{t|0,1} \tag{30}$$

*Proof.* Initially, let us deal with the dynamic objective in Eq. (29)

$$\min_{p_t} \min_{u_t} \int_0^1 \mathbb{E}_{p_t} \Big[ \frac{1}{2} \|u\|^2 + \mathbb{E}_{p_{0|t}} \big[ |u_t^\mathsf{T} \nabla G_{t,0}| + \frac{\sigma^2}{2} \Delta G_{t,0} \big] \Big] \mathrm{d}t \tag{31}$$

The marginal is constructed as a mixture of conditional probability paths $p_t = \int p_{t|0,1} \pi_{0,1} \mathrm{dx}_0 \mathrm{dx}_1$. Now, we sample pairs of $x_0, x_1$, and fix the couplings. This is equivalent to fixing the vector, which induces the couplings. Consequently, fixing the coupling results in recasting Eq. (29) as

$$\min_{p_{t|0,1}} \int \mathbb{E}_{p_t} \Big[ \frac{1}{2} \|u_{t|0,1}\|^2 + \big[ |u_{t|0,1}^\mathsf{T} \nabla G_{t|0}| + \frac{\sigma^2}{2} \Delta G_{t|0} \big] \Big] \mathrm{d}t \tag{32}$$

where $u_{t|0,1}$ is used to emphasize the drift that corresponds to the conditional probability path, $G_{t|0} \equiv G(X_t, x_0)$. Additionally, notice that by fixing the coupling, the expectation $\mathbb{E}_{p_{0|t}}$ is dropped.

$$\min_{p_t} \min_{X_t} \mathbb{E}_{p_{t_i|0,1}} \sum_{i=0}^N G(X_{t_i}, X_{t_{i+1}}) \tag{33}$$

Now, we want to prove that the conditional path preserves the marginal. Our process to show that is to isolate from every term in the Fokker Plank the joint distribution $\pi_{0,1}$, and preserve the Fokker Plank formulation after freezing the coupling.

- $\frac{\partial p_t}{\partial t} = \frac{\partial}{\partial t} \int p_{t|0,1} \pi_{0,1} \mathrm{dx}_0 \mathrm{dx}_1 = \int \frac{\partial \mathrm{p}_{t|0,1}}{\partial t} \pi_{0,1} \mathrm{dx}_0 \mathrm{dx}_1 = \mathbb{E}_{\pi_{0,1}} \big[ \frac{\partial \mathrm{p}_{t|0,1}}{\partial t} \big]$

- $\nabla \cdot (u_{t|0,1} p_t) = \nabla \cdot (u_{t|0,1} \int p_{t|0,1} \pi_{0,1} \mathrm{dx}_0 \mathrm{dx}_1) = \int \nabla \cdot (\mathrm{u}_{t|0,1} \mathrm{p}_{t|0,1}) \pi_{0,1} \mathrm{dx}_0 \mathrm{dx}_1 = \mathbb{E}_{\pi_{0,1}} [\nabla \cdot (\mathrm{u}_{t|0,1} \mathrm{p}_{t|0,1})]$

- $\Delta p_t = \nabla \cdot \nabla p_t = \nabla \cdot \nabla \int p_{t|0,1} \pi_{0,1} \mathrm{dx}_0 \mathrm{dx}_1 = \int \nabla \cdot \nabla \mathrm{p}_{t|0,1} \pi_{0,1} \mathrm{dx}_0 \mathrm{dx}_1 = \mathbb{E}_{\pi_{0,1}} [\Delta \mathrm{p}_{t|0,1}]$

where in every term we used Fubini's Theorem, to change the order between the integral from the definition of the marginal and the differentations from the FPE. Therefore, we can conclude that by freezing the coupling the Eq. (29) is recasted as

$$\min_{p_{t|0,1}} \int \mathbb{E}_{p_t} \Big[ \frac{1}{2} \|u_{t|0,1}\|^2 + \big[ |u_{t|0,1}^\mathsf{T} \nabla G_{t|0}| + \frac{\sigma^2}{2} \Delta G_{t|0} \big] \Big],$$
$$\text{s.t. } \frac{\partial p_{t|0,1}}{\partial t} = -\nabla \cdot (u_{t|0,1} p_{t|0,1}) + \frac{\sigma^2}{2} \Delta p_{t|0,1} \tag{34}$$

$\square$

### A.3 PROOF OF PROPOSITION 3.5

*The convex conjugate of the Lagrangian in Eq. (10) is defined as the Hamiltonian $H(x_t, a_t, t) = \sup_{u_{t|0,1}} \langle u_{t|0,1}, a_t \rangle - L(x_t, u_{t|0,1}, t)$. The optimization with respect to the drift $u_{t|0,1}$ yields $u_{t|0,1} = a_t - g(a_t^\mathsf{T} \nabla G_{t|0}) \nabla G_{t|0}$, where $\nabla G_{t|0} \equiv G(X_t, x_0)$, and*

$$g(a_t^\mathsf{T} \nabla G_{t|0}) = \begin{cases} \frac{a_t^\mathsf{T} \nabla G_{t|0}}{\|\nabla G_{t|0}\|^2} & \text{if} \quad |a_t^\mathsf{T} \nabla G_{t|0}| \le \|\nabla G_{t|0}\|^2 \\ sgn(a_t^\mathsf{T} \nabla G_{t|0}) & \text{else} \end{cases}$$

*Finally, we obtain the Hamiltonian associated with the Lagrangian in Eq. (10)*

$$H(x_t, a_t, t) = \frac{1}{2} \|a_t - g(a_t^\mathsf{T} \nabla G_{t|0}) \cdot \nabla G_{t|0}\|^2 - \frac{\sigma^2}{2} \Delta G(X_t, x_0) \tag{35}$$

*Proof.* We begin with the definition of the Hamiltonian, dependent on the gradient field $a_t$.

$$H(x_t, a_t, t) = \sup_{u_{t|0,1}} \langle u_{t|0,1}, a_t \rangle - L(x_t, u_{t|0,1}, t) \tag{36}$$

A more detailed analysis on the gradient field is given in Neklyudov et al. (2023). Given the convexity of Eq. (10), we compute the Hamiltonian through the subgradient of the absolute value $\partial(|u_t^\mathsf{T}\nabla G|)$. Optimizing with respect to the conditioned drift $u_{t|0,1}$ yields the following

$$0 = a_t - (u_t + \partial(u_t^\mathsf{T}\nabla G_t)(\nabla G_t)) \Rightarrow u_{t|0,1}^\star = a_t - \partial(u_t^\mathsf{T}\nabla G_t) \cdot (\nabla G_t) \tag{37}$$

where $\partial(\cdot)$ denotes the subgradient. Note that

$$\begin{cases} u_t^\mathsf{T}\nabla G_t > 0 : u_t = a_t - (\nabla G_t) \Rightarrow (a_t - (\nabla G_t))^\mathsf{T}\nabla G_t \geq 0 \Rightarrow a_t^\mathsf{T}\nabla G_t \geq \|\nabla G_t\|^2 \geq 0 \\ u_t^\mathsf{T}\nabla G_t < 0 : u_t = a_t + (\nabla G_t) \Rightarrow (a_t + (\nabla G_t))^\mathsf{T}\nabla G_t \leq 0 \Rightarrow a_t^\mathsf{T}\nabla G_t \leq -\|\nabla G_t\|^2 \leq 0 \\ u_t^\mathsf{T}\nabla G_t = 0 : \partial(u_t^\mathsf{T}\nabla G_t) = [-1,1], \text{ from which we infer } a_t^\mathsf{T}\nabla G_t = \alpha\|\nabla G_t\|^2, \text{ for } \alpha \in [-1,1]. \end{cases}$$

This essentially means that the inner products $u_t^\mathsf{T}\nabla G_t$, and $a_t^\mathsf{T}\nabla G_t$ share the same sign, namely $sgn(u_t^\mathsf{T}\nabla G_t) = sgn(a_t^\mathsf{T}\nabla G_t)$, when $u_t^\mathsf{T}\nabla G_t \neq 0$, otherwise $a_t^\mathsf{T}\nabla G_t = \alpha\|\nabla G_t\|^2$.

$$\begin{cases} \partial(u_t^\mathsf{T}\nabla G_t) = sgn(a_t^\mathsf{T}\nabla G_t), & \text{for} \quad |a_t^\mathsf{T}\nabla G_t| \geq \|\nabla G_t\|^2 \\ \partial(u_t^\mathsf{T}\nabla G_t) = \frac{a_t^\mathsf{T}\nabla G_t}{\nabla\|G_t\|^2}, & \text{for} \quad |a_t^\mathsf{T}\nabla G_t| \leq \|\nabla G_t\|^2 \end{cases}$$

We define function $g : \mathbb{R} \to \mathbb{R}$, to be equal to:

$$g(a_t^\mathsf{T}\nabla G_t) = \begin{cases} sgn(a_t^\mathsf{T}\nabla G_t), & \text{for} \quad |a_t^\mathsf{T}\nabla G_t| \geq \|\nabla G_t\|^2 \\ \frac{a_t^\mathsf{T}\nabla G_t}{\nabla\|G_t\|^2}, & \text{for} \quad |a_t^\mathsf{T}\nabla G_t| \leq \|\nabla G_t\|^2 \end{cases}$$

and substitute it in Eq. (36), which yields an expression for $u_t$ as a function of the gradient field

$$u_{t|0,1} = a_t - g(a_t^\mathsf{T}\nabla G_t) \cdot \nabla G_t \tag{38}$$

Now, we will provide an expression for the Hamiltonian dependent only on the gradient field $a_t$ and the feedback term $\nabla G_t(X_t, x_0)$.

$$H(x_t, a_t, t) = \langle a_t - g_t\nabla G_t, a_t\rangle - \left(\frac{1}{2}\|a_t - g_t\nabla G_t\|^2 - \langle a_t - g_t\nabla G_t, \nabla G_t\rangle\right) - \frac{\sigma^2}{2}\Delta G_t \tag{39}$$

Therefore, we conclude that the Hamiltonian can be expressed as:

$$H(x_t, a_t, t) = \frac{1}{2}\|a_t - g(a_t^\mathsf{T}\nabla G_t) \cdot \nabla G_t\|^2 - \frac{\sigma^2}{2}\Delta G_t \tag{40}$$

$\square$

## A.4 PROOF OF PROPOSITION 3.6

*Given the optimal conditional drift $u_{t|0,1}^\star$, we match the parameterized drift $u_t^\theta$, by minimizing the optimality gap given by our Entropic Lagrangian Bridge Matching objective as*

$$\min_\theta \int_0^1 \mathbb{E}_{p_{0,1}}\mathbb{E}_{p_{t|0,1}}\|a_t^\theta - u_{t|0,1}^\star - g(a_t^\mathsf{T}\nabla G_t) \cdot \nabla G_{t|0}\|^2 \mathrm{d}t \tag{41}$$

*From Proposition 3.5, we find that the parameterized drift can be expressed as $u_t^\theta = a_t^\theta - g(a_t^\mathsf{T}\nabla G_t) \cdot \nabla G_{t|0}$. Therefore, the matching objective of Eq. (13) can be rewritten as follows*

$$\min_\theta \int_0^1 \mathbb{E}_{p_t}\|u_{t|0,1}^\star - u_t^\theta\|^2 \mathrm{d}t \tag{42}$$

*Proof.* Given the prescribed path, we seek to learn a parameterized drift field that matches the optimal conditioned paths, acquired from the previous step, which minimized the Lagrangian in Eq. (30). Following advancements in matching frameworks, we extend the notion of bridging the variational gap for an entropy regularized optimal transport problem with general langriangian cost. We will first prove the following proposition which shows that we can express the variational gap that we need to bridge is expressed through the Bregman divergence.

**Proposition A.1.** *For a Lagrangian function $L(X_t, u_t, t)$, the convex conjugate is given by the Hamiltonian: $H(x_t, a_t, t) = \sup\langle a_t, u_t\rangle - L(x_t, u_t, t)$, dependent on the gradient field $a_t$. The variational gap between the optimal drift and the parameterized gradient field is expressed as the following Bregman divergence.*

$$\mathcal{L}^\theta - \mathcal{L}^\star = \int_0^1 \int_{\mathbb{R}^d} D_{L,H}[a_t : u_{t|0,1}^\star] p_{t|0,1} \mathrm{d}x \mathrm{d}t \tag{43}$$

*Proof.* Starting from a general minimization problem for a Lagrangian strictly convex with $u_t$

$$\min_{u_{t|0,1}} \int_0^1 \int_{\mathbb{R}^d} L(x_t, u_{t|0,1}, t) p_{t|0,1} \mathrm{d}x\mathrm{d}t, \text{ subj.: } \frac{\partial p_{t|0,1}}{\partial t} = -\nabla\cdot(p_{t|0,1}u_{t|0,1}) + \frac{1}{2}\sigma^2 \Delta p_{t|0,1}, \text{ and } p_0 = x_0, p_1 = x_1 \tag{44}$$

and expressing it as a Lagrange optimization problem yields, with $s_t$ is the Lagrange multiplier enforcing the Fokker Plank equation constraint

$$\begin{aligned}
\mathcal{L}(\theta) &= \min_{u_{t|0,1}} \int_0^1 \int_{\mathbb{R}^d} L(x_t, u_{t|0,1}, t) p_{t|0,1} \mathrm{d}x\mathrm{d}t, \quad \text{subj.: } \frac{\partial p_{t|0,1}}{\partial t} = -\nabla\cdot(p_{t|0,1}u_{t|0,1}) + \frac{1}{2}\sigma^2\Delta p_{t|0,1}\\
&= \sup_{s_t}\min_{u_{t|0,1}} \int_0^1 \int_{\mathbb{R}^d} L(x_t, u_{t|0,1}, t) p_{t|0,1}\mathrm{d}x\mathrm{d}t + \int_0^1 \int_{\mathbb{R}^d} s_t\left(\frac{\partial p_{t|0,1}}{\partial t} + \nabla\cdot(p_{t|0,1}u_{t|0,1}) - \frac{1}{2}\sigma^2\Delta p_{t|0,1}\right)\mathrm{d}x\mathrm{d}t\\
&= \sup_{s_t}\min_{u_{t|0,1}} \underbrace{\int_0^1 \int_{\mathbb{R}^d} L(x_t, u_{t|0,1}, t) p_{t|0,1}\mathrm{d}x\mathrm{d}t}_{I_1} + \underbrace{\int_0^1 \int_{\mathbb{R}^d} s_t\frac{\partial p_{t|0,1}}{\partial t}\mathrm{d}x\mathrm{d}t}_{I_2}\\
&\quad + \underbrace{\int_0^1 \int_{\mathbb{R}^d} s_t\nabla\cdot(p_{t|0,1}u_{t|0,1})\mathrm{d}x\mathrm{d}t}_{I_3} - \underbrace{\int_0^1 \int_{\mathbb{R}^d} \frac{s_t}{2}\sigma^2\Delta p_{t|0,1}\mathrm{d}x\mathrm{d}t}_{I_4}
\end{aligned} \tag{45}$$

We separate each one of the integrals and perfrom integration by parts

- $I_2 = \int_0^1 \int_{\mathbb{R}^d} s_t\frac{\partial p_{t|0,1}}{\partial t}\mathrm{d}x\mathrm{d}t = [s_t p_{t|0,1}]_0^1 - \int_0^1 \int_{\mathbb{R}^d} p_{t|0,1}\frac{\partial s_t}{\partial t}\mathrm{d}x\mathrm{d}t$

- $I_3 = \int_0^1 \int_{\mathbb{R}^d} s_t\nabla\cdot(p_{t|0,1}u_{t|0,1})\mathrm{d}x\mathrm{d}t = -\int_0^1 \int_{\mathbb{R}^d}\langle\nabla s_t, u\rangle p_{t|0,1}\mathrm{d}t\mathrm{d}x + \frac{\sigma^2}{2}\int_0^1 \oint s_t\langle\nabla p_{t|0,1}, d\mathbf{n}\rangle\mathrm{d}t$

- $I_4 = \oint s_t\langle\nabla p_{t|0,1}, d\mathbf{n}\rangle\mathrm{d}t - \int_0^1 \int_{\mathbb{R}^d} s_t\Delta p_{t|0,1}\mathrm{d}t\mathrm{d}x = \int_0^1 \int_{\mathbb{R}^d}\langle\nabla s_t, \nabla p_{t|0,1}\rangle\mathrm{d}x = -\int_0^1 \int_{\mathbb{R}^d}\Delta s_t\, p_{t|0,1}\mathrm{d}x + \frac{\sigma^2}{2}\oint s_t\langle p_{t|0,1}, d\mathbf{n}\rangle$

Substituting them back to Eq. (45), we obtain

$$\begin{aligned}
\mathcal{L} &= \sup_{s_t}\min_{u_{t|0,1}} \int_0^1 \int_{\mathbb{R}^d}\left(L(x_t, u_{t|0,1}, t) - \frac{\partial s_t}{\partial t} - \langle\nabla s_t, u\rangle - \Delta s_t\right) p_{t|0,1}\mathrm{d}x\mathrm{d}t - \mathbb{E}_{p_0}[x_0] + \mathbb{E}_{p_1}[x_1]\\
&\quad + \frac{\sigma^2}{2}\int\oint s_t\langle p_{t|0,1}, d\mathbf{n}\rangle\\
&= \sup_{s_t}\int_0^1 \int_{\mathbb{R}^d}\left(-H(x_t, \nabla s_t, t) - \frac{\partial s_t}{\partial t} - \Delta s_t\right) p_{t|0,1}\mathrm{d}x\mathrm{d}t - \mathbb{E}_{p_0}[x_0] + \mathbb{E}_{p_1}[x_1] + \frac{\sigma^2}{2}\int\oint s_t\langle p_{t|0,1}, d\mathbf{n}\rangle\\
&= \min_{s_t}\int_0^1 \int_{\mathbb{R}^d}\left(H(x_t, \nabla s_t, t) + \frac{\partial s_t}{\partial t} + \Delta s_t\right) p_{t|0,1}\mathrm{d}x\mathrm{d}t + \mathbb{E}_{p_0}[x_0] - \mathbb{E}_{p_1}[x_1] - \frac{\sigma^2}{2}\int\oint s_t\langle p_{t|0,1}, d\mathbf{n}\rangle
\end{aligned} \tag{46}$$

where, the second equation stems from defining the Hamiltonian of the Lagrangian $L(x_t, u_t, t)$ with respect to the Lagrange multiplier $s_t$ as: $H(x_t, \nabla s_t, t) = \sup\langle u_t, \nabla s_t\rangle - L(x_t, u_t, t)$. Now, consider

the optimality gap between a parameterized gradient field, and an optimal gradient field $s_t^\star$, which corresponds to a loss function $\mathcal{L}^\star$. We compute the optimality gap as

$$
\begin{aligned}
\mathcal{L}^\theta - \mathcal{L}^\star &= \int_0^1 \int_{\mathbb{R}^d} (H(x_t, \nabla s_t, t) + \frac{\partial s_t}{\partial t} + \Delta s_t)\, p_{t|0,1} \mathrm{d}x \mathrm{d}t + \mathbb{E}_{p_0}[x_0] - \mathbb{E}_{p_1}[x_1] - \frac{\sigma^2}{2} \int \oint s_t \langle p_{t|0,1}, d\mathbf{n} \rangle \\
&\quad - \int_0^1 \int_{\mathbb{R}^d} (H(x_t, \nabla s_t^\star, t) + \frac{\partial s_t^\star}{\partial t} + \Delta s_t^\star)\, p_{t|0,1} \mathrm{d}x \mathrm{d}t + \mathbb{E}_{p_0}[x_0] - \mathbb{E}_{p_1}[x_1] - \frac{\sigma^2}{2} \int \oint s_t^\star \langle p_{t|0,1}, d\mathbf{n} \rangle \\
&= \int_0^1 \int_{\mathbb{R}^d} (H(x_t, \nabla s_t, t) - H(x_t, \nabla s_t^\star, t)) p_{t|0,1} \mathrm{d}x \mathrm{d}t + \underbrace{\int_0^1 \int_{\mathbb{R}^d} \frac{\partial s_t}{\partial t} p_{t|0,1} \mathrm{d}x \mathrm{d}t + \mathbb{E}_{p_0}[x_0] - \mathbb{E}_{p_1}[x_1]}_{-\int_0^1 \int_{\mathbb{R}^d} \frac{\partial p_{t|0,1}}{\partial t} s_t \mathrm{d}x \mathrm{d}t} \\
&\quad \underbrace{- \int_0^1 \int_{\mathbb{R}^d} \frac{\partial s_t^\star}{\partial t} p_{t|0,1} \mathrm{d}x \mathrm{d}t + \mathbb{E}_{p_0}[x_0] - \mathbb{E}_{p_1}[x_1]}_{\int_0^1 \int_{\mathbb{R}^d} \frac{\partial p_{t|0,1}}{\partial t} s_t^\star \mathrm{d}x \mathrm{d}t} + \underbrace{\int_0^1 \int_{\mathbb{R}^d} \Delta s_t p_{t|0,1} \mathrm{d}x \mathrm{d}t - \frac{\sigma^2}{2} \int \oint s_t \langle p_{t|0,1}, d\mathbf{n} \rangle}_{\int_0^1 \int_{\mathbb{R}^d} \Delta p_{t|0,1} s_t \mathrm{d}x \mathrm{d}t} \\
&\quad \underbrace{- \int_0^1 \int_{\mathbb{R}^d} \Delta s_t^\star p_{t|0,1} \mathrm{d}x \mathrm{d}t - \frac{\sigma^2}{2} \int \oint s_t \langle p_{t|0,1}, d\mathbf{n} \rangle}_{\int_0^1 \int_{\mathbb{R}^d} \Delta p_{t|0,1} s_t^\star \mathrm{d}x \mathrm{d}t} \\
&= \int_0^1 \int_{\mathbb{R}^d} (H(x_t, \nabla s_t, t) - H(x_t, \nabla s_t^\star, t)) p_{t|0,1} \mathrm{d}x \mathrm{d}t + \int_0^1 \int_{\mathbb{R}^d} (s_t - s_t^\star)(-\frac{\partial p_{t|0,1}}{\partial t} + \Delta p_{t|0,1}) \mathrm{d}x \mathrm{d}t \\
&= \int_0^1 \int_{\mathbb{R}^d} (H(x_t, \nabla s_t, t) - H(x_t, \nabla s_t^\star, t)) p_{t|0,1} \mathrm{d}x \mathrm{d}t + \int_0^1 \int_{\mathbb{R}^d} (s_t - s_t^\star)(\nabla \cdot (p_{t|0,1} u_{t|0,1}^\star)) \mathrm{d}x \mathrm{d}t \\
&= \int_0^1 \int_{\mathbb{R}^d} (H(x_t, \nabla s_t, t) - H(x_t, \nabla s_t^\star, t)) - \langle \nabla(s_t - s_t^\star), u_{t|0,1}^\star \rangle p_{t|0,1} \mathrm{d}x \mathrm{d}t
\end{aligned}
$$
(47)

At this point, we define the gradient field function $a_t(X_t) : \mathbb{R}^d \to \mathbb{R}^d$, which is equal to $\nabla s_t = a_t$, which finally yields

$$
\begin{aligned}
\mathcal{L}^\theta - \mathcal{L}^\star &= \int_0^1 \int_{\mathbb{R}^d} H(x_t, a_t, t) + L(x_t, u_{t|0,1}^\star, t)) - \langle a_t, u_{t|0,1}^\star \rangle p_{t|0,1} \mathrm{d}x \mathrm{d}t \\
&= \int_0^1 \int_{\mathbb{R}^d} D_{L,H}[a_t : u_{t|0,1}^\star] p_{t|0,1} \mathrm{d}x \mathrm{d}t
\end{aligned}
$$
(48)

where the first equality stems from the fact that at the optimal point $(a_t^\star, u_t^\star)$, it holds that $H(x_t, a_t^\star, t) = \langle a_t^\star, u_t^\star \rangle - L(x_t, u_t^\star, t)$ (Neklyudov et al., 2023; Boyd & Vandenberghe, 2004). This allows us to substitute the expression in the last line of Eq. (47), $\langle a_t^\star, u_t^\star \rangle - H(x_t, a_t^\star, t)$ with $L(x_t, u_t^\star, t)$, and express the variational gap as the Bregman Divergence, which emerges by the definition in the last equality finishing the proof of the proposition. $\qquad\square$

Therefore, to minimize the optimality gap, we sample sufficiently many uncoupled pairs $(x_0, x_1)$, and parameterize the gradient field $a_t \to a_t^\theta$, and try to match the parameterized gradient field on the optimized drift $u_t^\star$. We express our matching objective an expectation over the sampled pairs of optimality gap derived from the previous proposition

$$
\min_\theta \mathbb{E}_{\pi_{0,1}}[\mathcal{L}^\theta - \mathcal{L}^\star] = \min_\theta \mathbb{E}_{\pi_{0,1}} \Big[ \int_0^1 \int_{\mathbb{R}^d} H(x_t, a_t^\theta, t) + L(x_t, u_{t|0,1}^\star, t) - \langle a_t^\theta, u_t^\star \rangle p_{t|0,1} \mathrm{d}x \mathrm{d}t \Big]
$$
(49)

Substituting for the expressions we have for the functions $L$, and $H$, we obtain

$$
\begin{aligned}
\mathbb{E}_{\pi_{0,1}}[\mathcal{L}^\theta - \mathcal{L}^\star] = \int_0^1 \int_{\mathbb{R}^d} \Big( &\frac{1}{2} \|a_t^\theta - g_t \cdot \nabla G_t\|^2 - \frac{\sigma^2}{2} \Delta G_t \\
&+ \frac{1}{2} \|u_{t|0,1}^\star\|^2 + |u_{t|0,1}^\star|^\intercal \nabla G_t + \frac{\sigma^2}{2} \Delta G_t - \langle a_t^\theta, u_{t|0,1}^\star \rangle \Big) p_{t|0,1} \pi_{0,1} \mathrm{d}x \mathrm{d}t
\end{aligned}
$$
(50)

At this point, we take cases for the different values of $g_t$. **First:** $sgn(a_t^\mathsf{T}\nabla G_t) = sgn(u_t^{\star,\mathsf{T}}\nabla G_t) > 0$, and $g(a_t^\mathsf{T}\nabla G_t) = 1$:

$$
\begin{aligned}
\mathbb{E}_{\pi_{0,1}}[\mathcal{L}^\theta - \mathcal{L}^\star] &= \int_0^1 \int_{\mathbb{R}^d} (\frac{1}{2}\|a_t^\theta - \nabla G_t\|^2 + \frac{1}{2}\|u_{t|0,1}^\star\|^2 + u_{t|0,1}^{\star\mathsf{T}}\nabla G_t - \langle a_t^\theta, u_{t|0,1}^\star\rangle)p_t dxdt \\
&= \int_0^1 \int_{\mathbb{R}^d} (\frac{1}{2}\|a_t^\theta - \nabla G_t\|^2 + \frac{1}{2}\|u_{t|0,1}^\star\|^2 - \langle u_{t|0,1}^\star, a_t^\theta - \nabla G_t\rangle)p_t dxdt \\
&= \int_0^1 \int_{\mathbb{R}^d} (\frac{1}{2}\|a_t^\theta - \nabla G_t - u_{t|0,1}^\star\|^2)p_t dxdt
\end{aligned}
\tag{51}
$$

**Second:** $sgn(a_t\nabla G_t) = sgn(u_t^\star\nabla G_t) < 0$, and $g(a_t^\mathsf{T}\nabla G_t) = -1$:

$$
\begin{aligned}
\mathbb{E}_{\pi_{0,1}}[\mathcal{L}^\theta - \mathcal{L}^\star] &= \int_0^1 \int_{\mathbb{R}^d} (\frac{1}{2}\|a_t^\theta + \nabla G_t\|^2 + \frac{1}{2}\|u_{t|0,1}^\star\|^2 - u_{t|0,1}^{\star\mathsf{T}}\nabla G_t - \langle a_t^\theta, u_{t|0,1}^\star\rangle)p_t dxdt \\
&= \int_0^1 \int_{\mathbb{R}^d} (\frac{1}{2}\|a_t^\theta + \nabla G_t\|^2 + \frac{1}{2}\|u_{t|0,1}^\star\|^2 - \langle u_{t|0,1}^\star, a_t^\theta + \nabla G_t\rangle)p_t dxdt \\
&= \int_0^1 \int_{\mathbb{R}^d} (\frac{1}{2}\|a_t^\theta + \nabla G_t - u_{t|0,1}^\star\|^2)p_t dxdt
\end{aligned}
\tag{52}
$$

**Third:** $(u_t^\star\nabla G_t) = 0$, , and $g(a_t^\mathsf{T}\nabla G_t) = \lambda$ , with $\lambda \in [-1, 1]$:

$$
\begin{aligned}
\mathbb{E}_{\pi_{0,1}}[\mathcal{L}^\theta - \mathcal{L}^\star] &= \int_0^1 \int_{\mathbb{R}^d} (\frac{1}{2}\|a_t^\theta - \lambda\nabla G_t\|^2 + \frac{1}{2}\|u_{t|0,1}^\star\|^2 - \langle a_t^\theta, u_{t|0,1}^\star\rangle)p_t dxdt \\
&= \int_0^1 \int_{\mathbb{R}^d} (\frac{1}{2}\|a_t^\theta - \lambda\nabla G_t\|^2 + \frac{1}{2}\|u_{t|0,1}^\star\|^2 + \lambda u_{t|0,1}^{\star\mathsf{T}}\nabla G_t - \langle a_t^\theta, u_{t|0,1}^\star\rangle)p_t dxdt \\
&= \int_0^1 \int_{\mathbb{R}^d} (\frac{1}{2}\|a_t^\theta - \nabla G_t\|^2 + \frac{1}{2}\|u_{t|0,1}^\star\|^2 - \langle u_{t|0,1}^\star, a_t^\theta + \lambda\nabla G_t\rangle)p_t dxdt \\
&= \int_0^1 \int_{\mathbb{R}^d} (\frac{1}{2}\|a_t^\theta - \lambda\nabla G_t - u_{t|0,1}^\star|^2)p_t dxdt
\end{aligned}
\tag{53}
$$

where the second equality comes from the fact that we added the quantity $\lambda u_{t|0,1}^\star\nabla G_t = 0$. Finally, we can succinctly express the optimality gap combining Eq. (51), and 52:

$$
\min_\theta \mathbb{E}_{\pi_{0,1}}[\mathcal{L}^\theta - \mathcal{L}^\star] = \min_\theta \int_0^1 \int_{\mathbb{R}^d} (\frac{1}{2}\|a_t^\theta - g(a_t^\theta, \nabla G_t)\nabla G_t - u_{t|0,1}^\star\|^2)p_t dxdt
\tag{54}
$$

But notice that by the definition of $u_t^\theta$ in Eq. (38), we have $u_t^\theta = a_t^\theta - g(a_t^\theta, \nabla G_{t_{|0}})\nabla G_{t_{|0}}$, hence the minimization problem of the variational gap in Eq. (54) is expressed as

$$
\min_\theta \int_0^1 \int_{\mathbb{R}^d} \frac{1}{2}\|u_{t|0,1}^\star - u_t^\theta\|^2 p_t dxdt
\tag{55}
$$

$\square$

# B  DISCUSSIONS

## B.1  GAUSSIAN PATH APPROXIMATION

Finally, following recent advancements in matching frameworks (Liu et al., 2024; Albergo et al., 2023), the conditioned probability path is approximated in a simulation-free manner, as a Gaussian path

$$
p_{t|0,1} \approx \mathcal{N}(I_{t|0,1}, \sigma_t \mathbf{I}_d)
\tag{56}
$$

, where the process is given from

$$
X_t = I_{t|0,1} + \sigma_t Z
$$

---

**Algorithm 2** Spline Optimization

---

1: **Input**: $x_0, x_1, \{X_{t_k}\}$ where $0 < t_1 < \cdots < t_K < 1$, and the pairs from the key-point (KP) set $\{x_0^n, x_1^n\}_{n=1}^N$, and obtain their trajectories.
2: Initialize $I_t, \sigma_t$
3: **repeat**
4:     Sample $X_t \sim \mathcal{N}(I_t, \sigma_t^2 I_d)$
5:     Evaluate the objective $G(X_t, x_0)$ using the KP pairs
6:     Estimate the conditional drift $u_t(X_t | x_0, x_1)$ Eq. (11)
7:     Take gradient step w.r.t control pts. $\{X_{t_k}, \sigma_{t_k}\}$
8: **until** converges
9: Return $p_{t|0,1}$ parametrized by optimized $I_t, \sigma_t$

---

,where $I_{t|0,1} \equiv I(t, x_0, x_1)$ is the interpolant function between the pinned endpoints, and $Z \sim \mathcal{N}(0, \mathbf{I}_d)$ is the Brownian motion. evaluate $I_t$, and $\sigma_t$ through the following equations for the vector field and the score function (Särkkä & Solin, 2019)

$$\partial_t X_t = \partial I_t + \frac{\partial_t \sigma_t}{\sigma_t}(X_t - I_t), \ \ \nabla \log p_t(X_t) = -\frac{1}{\sigma_t^2}(X_t - I_t)$$

Lastly, using these expressions, we obtain the following expression for the conditional drift

$$u_t(X_t | x_0, x_1) = \partial_t I_t + \frac{\partial_t \sigma_t}{\sigma_t}(X_t - I_t) - \frac{\nu^2}{2\sigma_t^2}(X_t - I_t)$$

where $\nu$ is given by the definition of $\sigma_t = \nu\sqrt{t(1-t)}$. We can see that substitution of $I_t := (1-t)x_0 + t x_1$ and $\sigma_t := \nu\sqrt{t(1-t)}$, to the conditional drift expression indeed yields the desired drift of the Brownian bridge $\frac{x_1 - X_t}{1-t}$.

**Spline optimization.** Efficient optimization is achieved by parameterizing $I_t, \sigma_t$ respectively as $d$- and 1-D splines using several knots (i.e., control points) sampled sparsely and uniformly along the time steps $0 < t_1 < ... < t_K < 1$: $X_{t_k} \in \mathbb{R}^d$ and $\sigma_{t_k} \in \mathbb{R}$. Among those knots, we fit the time-dependent interpolant function $I_t$, and the standard deviation $\sigma_t$

$$I_t := \text{Spline}(t; \ x_0, \{X_{t_k}\}, x_1) \qquad \sigma_t := \text{Spline}(t; \ \sigma_0=0, \{\sigma_{t_k}\}, \sigma_1=0). \tag{57}$$

Notice that the parameterization in Eq. (57) satisfies the boundary in Eq. (56), hence remains as a feasible solution to Eq. (10). The number of control points $K$ is much smaller than the discretization steps ($K \leq 30$ for all experiments). This significantly reduces the memory complexity, compared to prior methods caching entire discretized SDEs (e.g. (De Bortoli et al., 2021; Chen et al., 2021)).

We follow the algorithm proposed in (Liu et al., 2024), shown in Alg. 2, which, crucially, involves no simulation of an SDE. This is attributed to the utilization of independent samples from $p_{t|0,1}$, which are known in closed form. Finally, recall that since we solve Eq. (10) for each pair $(x_0, x_1) \sim p_{0,1}^\theta$ and later marginalize to construct $p_t$.

## C ADDITIONAL DETAILS ON THE EXPERIMENTS

In this section, we provide further information about the experiments run to validate the efficacy of our FSBM.

**Baselines** We compare the efficacy of our FSBM in the crowd navigation tasks against the state-of-the-art matching framework GSBM (Liu et al., 2024), with the default hyperparamters. For the opinion depolarization task, we compare against the GSBM, and the DeepGSB (Liu et al., 2022a) also run with their default hyperparameters. For the DeepGSB, we adopt the "actor-critic" parameterization which yields better performance. Finally, in the image translation experiments, we compare against with DSBM (Shi et al., 2023), and Light and Optimal SBM (LOSBM) (Korotin et al., 2023). In our experiments, DSBM was implemented by considering trivial guidance returning the analytic solution of non-guided Brownian bridges. All other frameworks were run using their official implementation, and default hyperparameters. All methods, including our FSBM, are implemented in PyTorch (Paszke et al., 2019).

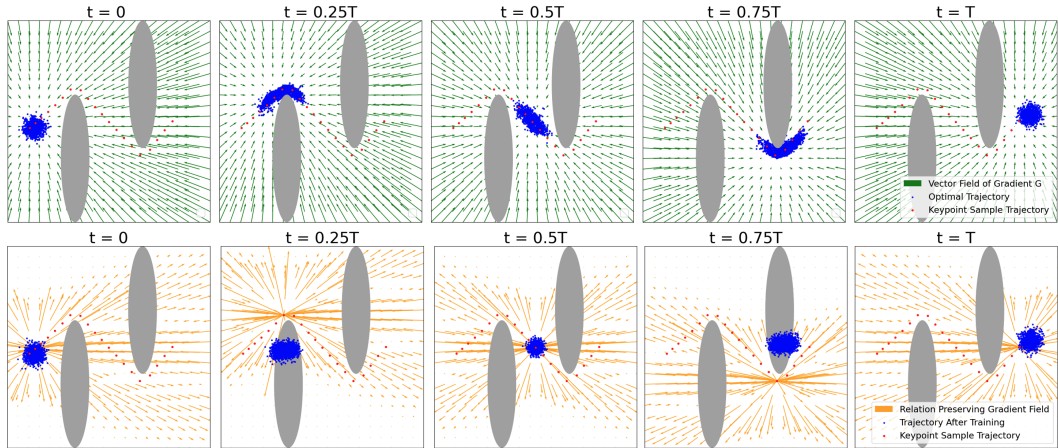

Figure 9: Comparison of gradient fields ensuing after training with UP: distance-preserving $G$, DOWN: relation-preserving $G$

**Guidance Function** Recall that our analysis begins with the static entropic OT problem in Eq. (3), to which an additional regularization term is included to capture the interaction among unpaired samples and aligned samples, thereby guiding the transport map. In practice, the selected Guidance Function was a slightly modified version of Eq. (7)

$$G(X_0, X_t) = \alpha \cdot (d(X_t, x_t^i) - d(X_0, x_0^i))^2 \tag{58}$$

where $\alpha \in \mathbb{R}$ is the guidance regularization hyperparameter responsible to amplify $G$, and bypass the minimization of the kinetic energy of the non-paired samples enough to satisfy the feasibility of the solution. where for our experiments, the selected distance between the non-coupled samples and the fixed KP sample was the $L_1$ norm $d(X_t, x_t^i) = \|X_t - x_t^i\|_{L1}$. The $i^{\text{th}}$ aligned data point was the closest keypoint sample to non-aligned particle $X_t$ at time $t = 0$. To compute the guidance function, we need the trajectories of the KP samples. These are assumed to be given only through their samples and fixed for each KP pair. In practice in the next sections, we discuss how we acquired those trajectories for each of our experiments.

**Relation-Preserving vs Distance Preserving** Although the relation-preserving scheme was indeed the more effective choice in the static setting in [1], this was not the case in our applications. This could be attributed to the fact that the softmax averaging relation preserving scheme works more effectively in static frameworks. However, in our dynamic formulation, we observed empirically that the softmax averaging in relation-preserving schemes (similar to [1]) failed to generate a gradient field capable of providing effective guidance to the unpaired samples. In contrast, the distance-preserving scheme offered significantly better guidance. Figure 9 presents a comparison of the ensuing gradient fields after training with the relation-preserving guidance function similar to (Gu et al., 2022), and our distance-preserving guidance function.

**Related Works (Semi-Supervised OT)** Recently, the field of Optimal Transport (OT) has seen advancements in developing partially supervised frameworks aimed at reducing the reliance on source-target aligned image pairs for training, thereby alleviating the high cost of labeling supervised datasets (Mustafa & Mantiuk, 2020). In particular, the semi-supervised domain adaptation framework (Sato et al., 2020) demonstrates how limited labeled target domain data can act as keypoints, enabling the adaptation of unlabeled data through iterative optimization. These approaches generally share a common theme: the way partial supervision is integrated into the OT formulation. For many applications, annotated keypoint pairs are pre-determined, and guidance is incorporated into the OT problem as regularization terms that constraint the transport map (Gu et al., 2023a). For example, (Courty et al., 2017) explored regularizing the OT cost using functions that encode class label information. Their regularization term was used to preserve the data structure while encouraging the alignment of labeled data with shared class labels across source and target domains for domain adaptation. Similarly, (Yan et al., 2018) proposed a method that uses the Gromov-Wasserstein (GW)

model to transport source samples to the target domain. Their approach induced constraints in the transport process by minimizing the distance between the centers of transported source samples and labeled target samples with the same class labels. Furthermore, (Lin et al., 2021) regularized the OT cost to capture the group structure between source and target distributions and their representative anchors. In this framework, the anchors promote clustering, similar to our approach, and are used to enhance robustness to outliers by imposing rank constraints on the transport plan. However, unlike the concept of keypoints in our approach, the anchors do not represent pairs of annotated points explicitly used to guide the transport map. More closely aligned with our approach, (Gu et al., 2023b) proposed a relation-preserving keypoint-guided model that steers OT matching by preserving both the correspondence between keypoint pairs and the relationships of each data point to the keypoints. In their framework, this guidance is introduced as a regularization term in the OT formulation. While the utilization of keypoitns by Gu et al. 2023 is similar to ours, their choice of the guidance function differs. Specifically, they modeled the distance as a softmax-averaged relationship between unlabeled samples and keypoints, with the guidance expressed as the Jensen-Shannon divergence between the softmax averages of the source and target distributions. Lastly, in (Gu et al., 2023a) it was shown that this partially supervised OT framework could be applied to effectively guide a score based diffusion model.

**Network Architectures**  For all experiments, we adopted the same architectures from GSBM, which consists of 4 to 5 residual blocks with sinusoidal time embedding. All networks are trained from scratch, without utilizing any pretrained checkpoint, and optimized with AdamW (Loshchilov, 2017)

**Forward and Backward Scheme**  Training of our FSBM entails a 'forward-backward' scheme originally proposed in DSBM (Shi et al., 2023). This necessitates the parameterization of two drifts, one for the forward SDE and another for the backward. During odd epochs, we simulate the coupling from the forward drift, solve the corresponding conditional lagrangian problem in Eq. (10), and then match the resulting $p_t$ with the backward drift. On the other hand, during even epochs, we perform the reverse procedure, matching the forward drift with the marginal $p_t$ derived from the backward drift. As discussed in Liu et al. (2024), this alternating scheme improves the performance of the matching algorithm, averting the matching framework from bias accumulation, as the forward drift always matches the terminal distribution $\pi_1$, and the backward drift is ensured to match the source distribution $\pi_0$.

## C.1 Crowd Navigation

We revisit the efficacy of our FSBM in solving complex crowd navigation tasks and its superior capability to generalize under a variety of initial conditions. More explicitly, the two compared frameworks were trained on transporting the samples $X_0 \sim \pi_0$ to $X_1 \sim \pi_1$. In the evaluation phase, the task is always to transport the samples to the same target distribution as in the training phase, however, the initial conditions change throughout the different tasks. For the first task (Vanilla), we draw new samples from the same distribution $\pi_0 = \mathcal{N}(\mu_0, \sigma_0)$, for the second task we shift the mean of the distribution, for the third task we increase the standard deviation, and in the fourth task we sample from a uniform distribution. The initial conditions for each task are the following:

- Vanilla: $\mathcal{N}\Big([-11, -1], 0.5\Big)$

- Perturbed Mean: $\mathcal{N}\Big([-11, -4], 0.5\Big)$

- Perturbed STD: $\mathcal{N}\Big([-11, -1], 3\Big)$

- Uniform Distribution with geometric mean $[-11, -1]$

where $\mathcal{N}([\mu_1, \mu_2], \sigma^2)$ represents a Gaussian with mean the point $[\mu_1, \mu_2]$, and standard deviation equal to $\sigma$. Note that for the S-tunnel crowd navigation, we used 100 KPs, which is approximately 4%, of the entire population, and in the V-neck task only 20 KP samples were employed, which is approximately 1% of the population. In our experiments, for the trajectories of the aligned data, we utilized the POT python package (Flamary et al., 2021), to acquire optimal pairings, and GSBM for one epoch to obtain the trajectories between the endpoints. Figures 10, and 11 illustrate the

trajectories of the coupled samples, that guide the matching of the unpaired particles. Finally, Figure 13 presents additional plots of FSBM in the crowd navigation tasks examined under the variety of initial conditions mentioned above.

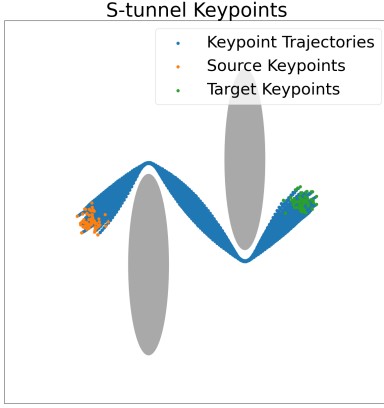
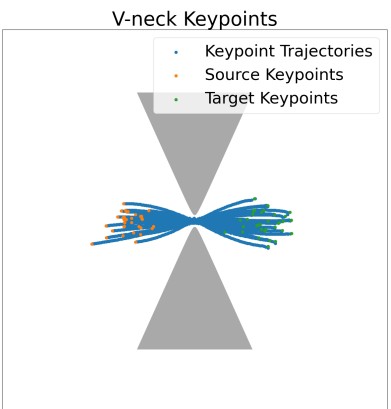

Figure 10: Interpolating path of the KP samples for the S-tunnel

Figure 11: Interpolating path of the KP samples for the V-neck

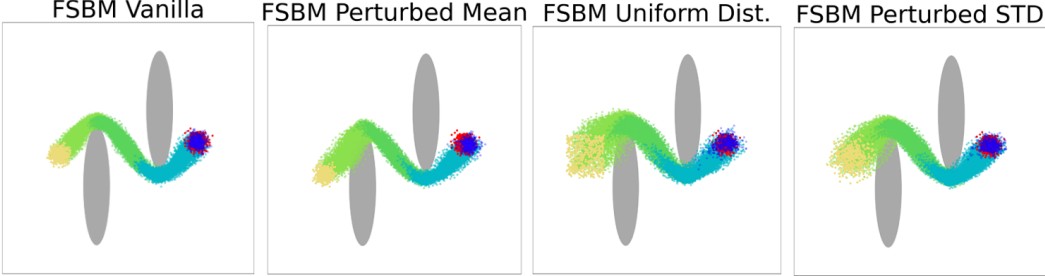

Figure 12: Additional Figures on S-tunnel using our FSBM. The color of particles means: **i) Yellow:** initial conditions, **ii) Green and Cyan:** intermediate trajectory, **iii) Red:** target distribution, **iv) Navy Blue:** generated distribution

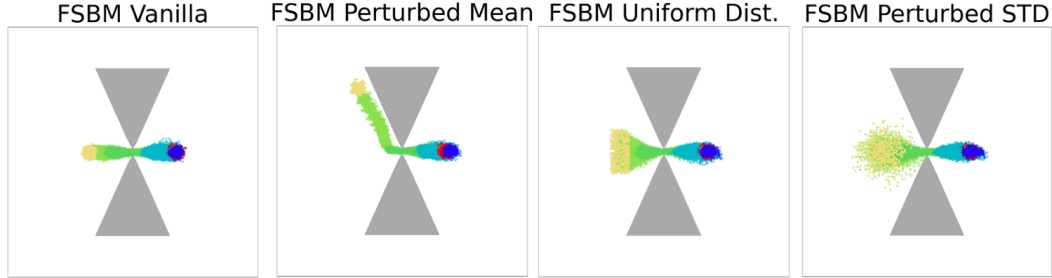

Figure 13: Additional Figures on V-neck using our FSBM. The color of particles means: **i) Yellow:** initial conditions, **ii) Green and Cyan:** intermediate trajectory, **iii) Red:** target distribution, **iv) Navy Blue:** generated distribution

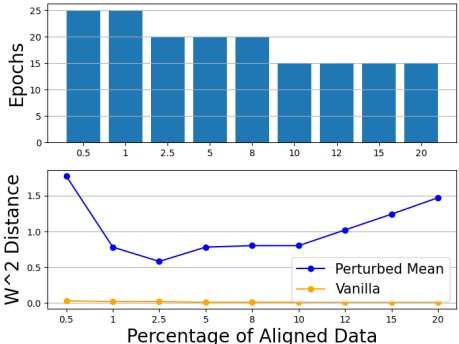

Figure 14: UP: Training Epochs, DOWN: The $W^2$ Distance under Vanilla and Perturbed Mean in the S-tunnel for varying the number of KP samples

Table 6: The $W^2$ Distance under Vanilla and Perturbed Mean in the S-tunnel for varying the number of KP samples

| Percentage of KPs | $\mathcal{W}^2$ Vanilla | $\mathcal{W}^2$ Pert. Mean |
|---|---|---|
| 1 | 0.034 | 0.77 |
| 2.5 | 0.027 | 0.55 |
| 5 | 0.018 | 0.79 |
| 8 | 0.015 | 0.82 |
| 10 | 0.013 | 0.81 |
| 12 | 0.010 | 1.01 |
| 15 | 0.008 | 1.23 |
| 20 | 0.008 | 1.48 |

### C.1.1 IMPACT OF KP PAIRS IN THE S-TUNNEL

In our methodology, it becomes clear that the number of KP pairs used to guide the rest of the samples affects the effectiveness of our algorithm. In Fig. 14, and Table **??**, we see that the vanilla performance of our framework continues to increase as the number of aligned pairs increases. As depicted in Figure 14a, although an increase of the aligned pairs results in reduced training time. Additionally, in Fig. 14, we see that the vanilla performance of our framework continues to increase as the number of aligned pairs increases. However, it is shown that the empirical generalization to unseen initial conditions (e.g. when the mean of the initial distribution is perturbed) does not necessarily improve by adding more keypoints, suggesting an overfitting like phenomenon. There appears to be a "sweet spot", which suggests that using too many KP pairs hinders the ability of our algorithm to generalize effectively akin to an overfitting-like effect. Lastly, it is important to note that the hyperparameters of the framework were tuned to maximize the performance while using few keypoints. As a result, increasing the number of aligned pairs may require re-tuning the hyperparameters to maintain optimal performance. Lastly, it should be mentioned that even this decrease in generalizability is still very small compared to the benchmark methods. Deeper understanding the effect of the aligned data points in the efficacy, and generalizability of the model is a topic for future work.

### C.2 OPINION DYNAMICS

Subsequently, we revisit the high-dimensional opinion depolarization of a population. Each particle possesses a high-dimensional opinion $X_t \in \mathbb{R}^{100}$ that evolves under polarizing dynamics

$$dX_t = f_{\text{polarize}}(X_t, t)dt + \sigma dW_t \tag{59}$$

which tend to segregate them into two groups of diametrically opposing opinions.

**Polarization drift**  We use the same polarization drift from DeepGSB and GSBM Liu et al. (2022a; 2024), based on the party model (Gaitonde et al., 2021). At each time step $t$, all agents receive the same random information $\xi_t \in \mathbb{R}^d$ sampled independently of $p_t$, then react to this information according to

$$f(x; p_t, \xi_t) := \mathbb{E}_{y \sim p_t}[a(x, y, \xi_t)\bar{y}], \quad a(x, y, \xi_t) := \begin{cases} 1 & \text{if } \text{sign}(\langle x, \xi_t \rangle) = \text{sign}(\langle y, \xi_t \rangle) \\ -1 & \text{otherwise} \end{cases}, \tag{60}$$

where $x, y$ are opinions and $\bar{y} := \frac{y}{\|y\|^{\frac{1}{2}}}$ and $a(x, y, \xi_t)$ is the agreement function indicating whether the two opinions agree on the information $\xi_t$. This suggests that the agents tend to reject opinions, with which they disagree, while they accept opinions closer to theirs. As the dynamics of this model evolve

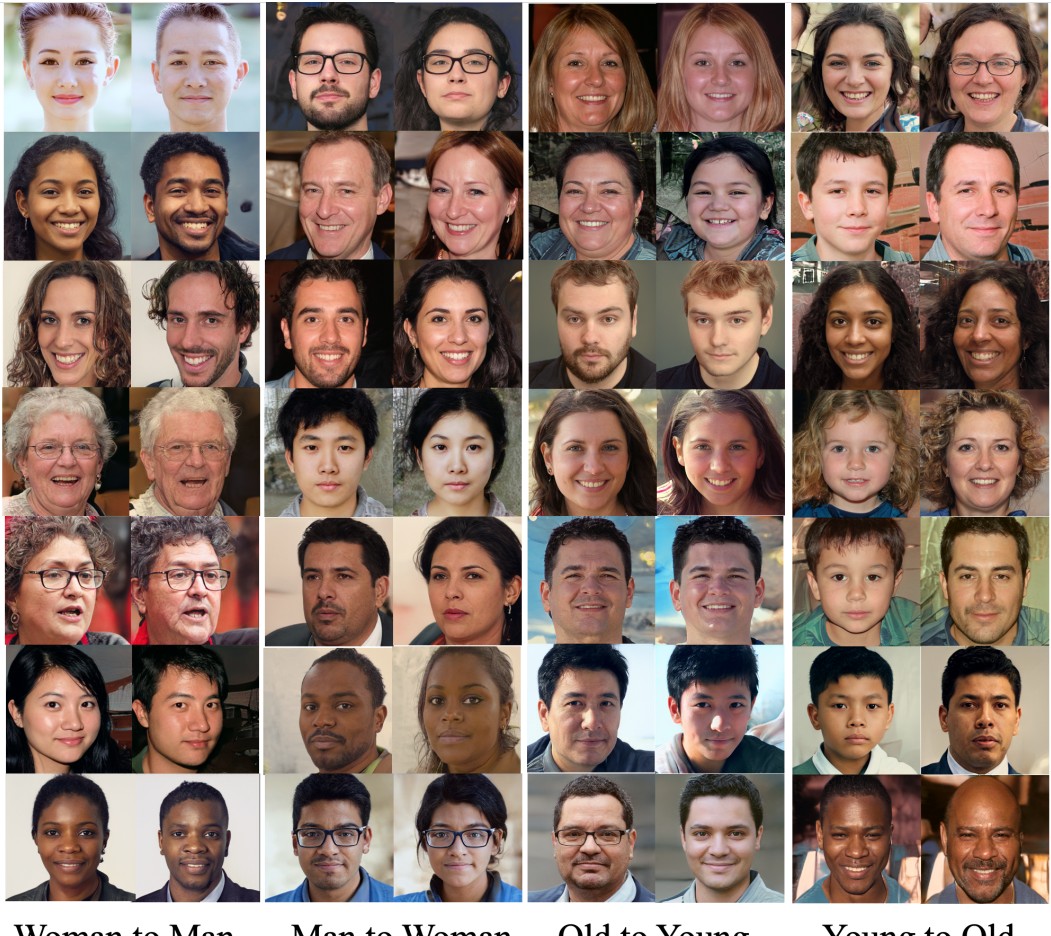

Woman to Man     Man to Woman     Old to Young     Young to Old

Figure 15: Additional Translation Examples

Recall that the size of the key-point set utilized for the opinion depolarization was approximately 2%, of the entire population. In our opinion depolarization experiments, the optimal pairings were obtained through the POT python package, however, in contrast to the crowd navigation tasks, in this case, the trajectories of the opinion guide particles were obtained by linear interpolating between initial and target positions.

## C.3    IMAGE TRANSLATIONS

Finally, we review the experiments in the two types of image translations: **i)** Gender translation, **ii)** Age translation. We follow the setup of (Gushchin et al. (2024)) with the pre-trained ALAE autoencoder (Pidhorskyi et al. (2020)) on 1024 ×1024 FFHQ dataset (Karras et al. (2019)) to perform the translation in the latent space of dimension $512 \times 1$ enable more efficient training and sampling.

Recall that the number of aligned images was 4% of the total number of images for the gender translation and 8% for the age translation. In our translation experiments, the aligned pairings, used to guide the rest of the matching algorithm, were obtained by employing a lightweight SB algorithm using Gaussian mixture parameterization (Korotin et al., 2023). Theoretically, this algorithm provides guarantees to solve the SB, and empirically the quality of the images generated in the training set of this algorithm was deemed satisfactory to constitute our KP set. Tables 7 and 11 present additional results establishing the superiority of the coupling obtained by our FSBM. More specifically, Table 7 presents the $L2$-norm of the difference between the pixel values of the input images and the generated images for each of the 4 translation tasks. Note that the values have also been divided by the total number of pixels ($1024 \times 1024$) to derive a mean $L2$-norm difference. Furthermore, Table 11 presents

the Structural Similarity Index Measure (SSIM) between the input and the generated images in the 4 translation tasks. Finally, Figure 15 presents additional translation instances using our FSBM.

Table 7: $L2 - \text{norm}/(1024 \times 1024)$ values between input and generated images in 4 translation tasks for DSBM, LOSBM, and our FSBM (Lower is better). Note the values are divided by $2^20$, which is the total number of pixels to get a mean $L2$-norm over all the pixels

| Optimizer | Man to Woman | Woman to Man | Old to Young | Young to Old |
|---|---|---|---|---|
| DSBM | 0.87 | 0.84 | 0.88 | **0.87** |
| LOSBM | 1.42 | 1.47 | 1.41 | 1.35 |
| FSBM | **0.79** | **0.81** | **0.83** | 0.88 |

Table 8: SSIM values between input and generated images in 4 translation tasks for DSBM, LOSBM, and our FSBM (Higher is better)

| Optimizer | Man to Woman | Woman to Man | Old to Young | Young to Old |
|---|---|---|---|---|
| DSBM | 0.46 | **0.51** | **0.50** | 0.43 |
| LOSBM | 0.34 | 0.35 | 0.35 | 0.36 |
| FSBM | **0.49** | 0.47 | 0.46 | **0.45** |

Finally, we also used ChatGPT to assess the image couplings between 100 pairs of input and generated images in each translation task. Given that the coupling from LOSBM was considerably worse than the coupling generated by the other two methods, we only performed comparison between DSBM [1] and our FSBM. ChatGPT was instructed to give a score on the coupling quality on a scale 0-5, with 5 being the best. The coupling quality was assessed using the following 3 criteria: i) Background Consistency: consistency in terms of color, lighting, background objects, ii) Identity Preservation: maintain the identity of the original face in terms of facial features and overall resemblance iii) Structural Integrity: preserve structural integrity of the input image in terms of accessories worn, pose, orientation. The results in 4 translation tasks are shown in the Tables below.

Table 9: Background Consistency score between input and generated images in 4 translation tasks for DSBM, and our FSBM (Higher is better), assigned by ChatGPT

| Optimizer | Man to Woman | Woman to Man | Old to Young | Young to Old |
|---|---|---|---|---|
| DSBM | 4.05 | 4.04 | 3.99 | 4.08 |
| FSBM | **4.58** | **4.21** | **4.05** | **4.10** |

Table 10: Identity Preservation score between input and generated images in 4 translation tasks for DSBM, and our FSBM (Higher is better), assigned by ChatGPT

| Optimizer | Man to Woman | Woman to Man | Old to Young | Young to Old |
|---|---|---|---|---|
| DSBM | 3.89 | 3.09 | 3.15 | 3.49 |
| FSBM | **4.02** | **3.48** | **3.72** | **3.60** |

Table 11: Structural Integrity score between input and generated images in 4 translation tasks for DSBM, and our FSBM (Higher is better), assigned by ChatGPT

| Optimizer | Man to Woman | Woman to Man | Old to Young | Young to Old |
|-----------|--------------|--------------|--------------|--------------|
| DSBM | 3.38 | 3.27 | 3.11 | 3.30 |
| FSBM | **3.69** | **3.61** | **3.55** | **3.33** |

