# OpenReview forum: "Feedback Schrödinger Bridge Matching"
_ICLR.cc/2025/Conference — ICLR 2025 Oral_

### Official Review · Reviewer_LqYG · 2024-10-31

**Soundness:** 2
**Presentation:** 3
**Contribution:** 3
**Rating:** 8
**Confidence:** 5

**Summary:**

This paper proposed a novel approach, which introduces Feedback Schrodinger Bridge Matching (FSBM) as a semi-supervised matching framework for addressing distribution transfer issues. The authors initiate their study with the static semi-supervised Optimal Transport (OT) problem. They judiciously incorporate an additional regularization term to capture the interaction between uncoupled and aligned samples, thereby enhancing the transfer mapping process.

The paper's strength lies in its innovative use of a state feedback term to encode information from aligned samples, effectively guiding uncoupled samples' transfer. Additionally, the authors optimize the drift field using divergence, supported by a variational objective.

Starting from the Static Euclidean OT problem, the proposed term $G$ is seamlessly integrated into the cost function $c$, and feasible optimization algorithms are derived using Lagrangian and Hamiltonian formulations. Overall, the paper's motivation, methodology, and experimental results are partially well-presented and contribute to the existing literature.

**Strengths:**

+ Introducing Feedback Schrödinger Bridge Matching (FSBM): FSBM is the first matching framework capable of leveraging information on partially aligned data pairs, significantly improving efficiency.
+ Semi-supervised guidance: Accelerate training and enhance generalization capabilities by using a small number of pre-aligned pairs as state feedback to guide the transfer mapping of uncoupled samples.
+ Dynamic Target Rewriting: Reformulate the static entropic optimal transport (EOT) problem into a dynamic form to exploit the scalability of the matching framework.
+ Experimental validation: Extensive experiments show that FSBM has better generalization ability under various unseen initial conditions and is faster in training time.
+ Multiple task applications: FSBM performs well in various matching tasks, such as low-dimensional crowd navigation, high-dimensional opinion depolarization, and image translation.

In particular, I think this paper's theoretical derivation and experiments are good, and utilizing partially paired samples is also novel and exciting.

**Weaknesses:**

1. The key contribution is the proposal of the distance function $G$. However, the motivation of $G$ is not clearly illustrated. Can authors further illustrate why they give such a constraint to paired samples? What is the meaning of $G$? What is the meaning of $d(X_1,x_1^i)$ and $d(X_0,x_0^i)$? Is it relative to clustering? Authors can illustrate the behaviors of paired samples and unpaired samples more specifically. Is the behavior of FSBM related to the "clustering" of trajectories? More discussions, examples, and theoretical derivations to illuminate the motivation of the term $G$ are expected to be shown in the paper.

2. Paired samples and stochastic properties of SDE. As we know, randomness is an important property of SDE/Diffusion/Bridges. From $x_0$ to $x_1$, if $x_0$ is sampled from $\pi_0$, $x_1$ can be stocastic. However, paired samples seem to be deterministic. What is the relation between paired samples and the stochastic property of SDE/Diffusion/Bridges? Please address this concern explicitly in the methodology section.

3. Figure 3 directly shows the effect of $G$. However, Figure 3 is qualitative and not rigorous. The term $u^{\top}\nabla{G}$ forces the trajectory of unpaired samples to the trajectory of paired samples. What is the reason? Please provide a more rigorous analysis of how G affects the trajectories, perhaps through a theoretical derivation or a quantitative experiment that demonstrates the effect.

4. The proof Theorem 3.2. In Eq. 20, authors discretize $G(X_0,X_1)$ into a sum sequence $G(X_{t_i},X_{t_{i+1}})$. The authors seem to assume $\delta t \rightarrow 0$ and transform the sequence of $G$ into $G(X_0, X_1)$. Is the application rigorous? Authors and other reviewers can confirm that. Please discuss under which conditions the transformation (i.e., from the sum sequence to the time interval $[0,1]$) is valid. I suggest that authors provide a proof sketch to make the paper more readable.

5. The contradiction of formulations of Hamiltonian function. As for the formulation of Hamiltonian function, in line 860, authors use $\sup_{u} \langle u,\nabla{a} \rangle-\mathcal{L}(x,u,t)$. However, in Eq. 37, authors $\sup_{u} \langle u,a\rangle-\mathcal{L}(x,u,t)$, which is distinct from line 860. In the following derivations, authors use the formulation in Eq. 37, where the Lagrangian multiplier $a$ is not with the gradient operator. However, in Eq. 47 of Proposition A.1, from line 998 to line 1000, authors eliminate $\langle  \nabla{a^*, u^*}\rangle$ using the product term in Hamiltonian function $H^*(x,a^*,t)$. Then, the authors use the Hamiltonian formulation with $\nabla{a^*}$. Thus, in the proof of Proposition 3.5 and the proof of Proposition A.1, the formulations of Hamiltonian functions are contradictory. I am not familiar with Hamiltonian function, but the contradiction in the proofs should be claimed.

**Questions:**

Minor issues:
1. The resolution of figures should be improved.
2. Some typos.
3. What are the advantages of the current FSBM framework's dynamic target design through state feedback? Is there any theoretical evidence that makes it more transparent?

---

> ### Author Response · Authors · 2024-11-20
> **Author Response to Reviewer LqYG (part 1/2)**
>
> We would like to thank the reviewer for their positive remarks, as well as their useful comments and further suggestions for improving our work. In the following we address each of the concerns and suggestions of the reviewer.
>
> ### **1. Motivation and Meaning of guidance function $G$**
> - The key contribution of our work was the derivation of a more general dynamic formulation. Our novelty lies in the fact that our analysis starts from a static regularized EOT problem (Eq. (8)), and derive a dynamic formulation (Eq. (9)). Importantly, we highlight that our methodology can be adapted to any class of differentiable regularizing functions, not just guidance regularizing function. Our approach bridges between more general static and dynamic problems, enabling us to harness the flexibility of adding regularization terms in static EOT formulation, and then acquire a more complete description of the transport and leverage recent advancements in matching frameworks.
> - The intuition behind the utilization of the guidance function $G$ in our approach is to capture structural information from the source distribution and propagate them throughout the entire trajectory. Importantly, the conditioning on the unpaired sample $x_0$ acts as an anchor. Our guidance function $G$ groups the unpaired data into clusters around the KPs in the source distribution, and penalize the deviation of the unpaired samples from the trajectory that their assigned keypoint sample follows, preserving structural and distance information in the dynamic transport map.  This is performed by evaluating the distances $d(X_0, x_0^i)$ which corresponds to the distance of an uncoupled sample $X_0$ to the $i^\text{th}$  keypoint $x_0^i$, and similarly the distance $d(X_t, x_t^i)$ for every time step $t\in [0,1].$ This forces unpaired samples to preserve their relative distance from the closest keypoint sample in the source distribution at each time-step.
> - Regularization using a guidance function has been implemented in previous works within the realm of semi-supervised Optimal Transport. These approaches leverage the guidance function $G$ to encode information about the relationship between unpaired data and keypoints. For example, in [1] the authors investigated incorporating various distinct $G$ functions into the Optimal Transport framework as regularization terms. They proposed a relation-preserving keypoint-guided model that steers OT matching by preserving both the correspondence between keypoint pairs and the relationships of each data point to the keypoints. While the utilization of keypoints [1]  is similar to our methodology, the nature of the guidance function differs. In our approach, the guidance function acts in a distance preserving manner. More specifically, $G$ creates clusters of unpaired points around their closest KeyPoint in the source distribution, and penalize the unpaired samples when they deviate from the trajectory that their assigned keypoint sample follows. This contributes to preserving structural information in the dynamic transport map.  A pertinent passage has been added in the appendix of the manuscript presenting more examples and illuminating the motivation behind the regularization with the guidance function $G.$
>
> ### **2. Stochastic Nature of Keypoint coupling**
> The guidance function is constructed deterministically with respect to the keypoints, similar to approaches seen in prior works [1, 2, 3]. However, this does not compromise the overall stochastic nature of the keypoints, as they are solutions of the static Schrodinger Bridge, which is inherently stochastic. In other words, sampling the keypoints from the static SB ensures that randomness is a fundamental component of the solution, rendering them valid for our guidance term. In practice, for our experiments we acquired the optimal pairings by sampling from the prior stochastic frameworks that approximate the Schrodinger Bridge (e.g., GSBM [4], LOSBM[5]). Given the inherent stochasticity of the static SB, there may be extreme instances where high stochasticity result in the keypoints coupling not providing with informative guidance. In such cases, we can easily reduce the influence of the guidance by adjusting its weighting. A pertinent comment on the stochasticity of the coupling of the keypoints has been added in the methodology section of the paper (line 183).
>
> ### **3. Update of Figure 3**
> We would like to thank the reviewer for their suggestion to improve Figure 3, in order to better explain the effect of $G$ on the guidance of the trajectories. We substituted the qualitative Figure 3 with a more rigorous representation of snapshots of the vector field of $\nabla G$ at times $t = \{0, \ 0.33T, \ 0.67T, \ T\}$. In the new Figure, it is evident that the gradient of the guidance function pushes the non-coupled data towards the trajectory of the KeyPoints.

---

> ### Author Response · Authors · 2024-11-20
> **Author Response to Reviewer LqYG (part 2/2)**
>
> ### **4. Dynamic Formulation**
> Our approach reinterprets the static modified EOT problem in Eq. (8) of the manuscript through the prism of entropic interpolation [6], the solution of dynamical OT formulation corresponds to solving ‘local’ versions of optimal transport problems between infinitesimally close distributions [7].  Starting from the EOT problem, from $t_0=0$ and $t_N=1$, we can consider any intermediate point $t_1$ and find the optimal paths $p_1^\star$, $p_2^\star$ between the endpoints in the two newly formed subproblems (i.e., from $t_0$ to $t_1$, and $t_1 $ to $t_N$). The union of the optimal subpaths forms the optimal policy of the initial problem, and the converse is also true. This can be extended for any number of intermediate points (which is equivalent to taking $\delta t\rightarrow 0), leading us to to consider ‘local’ versions of optimal transport problems between infinitesimally close distributions [7]. We have also provided a sketch of proof for Theorem 3.2 to enhance readibility.
>
> ### **5. Contradiction in the Hamiltonian**
> We would like to thank the reviewer, for identifying these contradictions attributed to typos and error in the notation in the proof of Proposition A.1. More specifically, the Lagrange multiplier in Eq. (45) is not the same as the vector field $a_t $ of the Hamiltonian $H(x_t, a_t, t)$, but rather for Lagrange multiplier $s_t$ it holds that $a_t = \nabla s_t$. We have corrected the notation errors and typos and further clarified the proof of the Proposition A.1.
>
> ### **6. Advantages of current FSBM framework**
> We emphasize that the dynamic formulation of our FSBM framework offers significant advantages, in terms of faster training and improved generalization. These benefits, stemming from leveraging the guidance of coupled pairs, were observed primarily through extensive empirical analysis.  The faster training is intuitively explained by the use of aligned pair information to guide the matching of non-aligned data. While there is no theoretical evidence to verify our empirical findings yet, we aim to address this in future work by deriving theoretical bounds for our FSBM, proving that the utilization of keypoints results in a transport map which is better capable at generalizing.
>
> ---
> ### References
>
> 1. Xiang Gu, Yucheng Yang, Wei Zeng, Jian Sun, and Zongben Xu. Keypoint-guided optimal transport,
> 2023b
> 2. Xiang Gu, Liwei Yang, Jian Sun, and Zongben Xu. Optimal transport-guided conditional score-based
> diffusion model. Advances in Neural Information Processing Systems, 36:36540–36552, 2023a
> 3. Chi-Heng Lin, Mehdi Azabou, and Eva L Dyer. Making transport more robust and interpretable by
> moving data through a small number of anchor points. Proceedings of machine learning research,
> 139:6631, 2021
> 4. Guan-Horng Liu, Yaron Lipman, Maximilian Nickel, Brian Karrer, Evangelos A Theodorou, and
> Ricky TQ Chen. Generalized Schrödinger bridge matching. In International Conference on
> Learning Representations, 2024.
> 5. Nikita Gushchin, Sergei Kholkin, Evgeny Burnaev, and Alexander Korotin. Light and optimal
> schrödinger bridge matching. In Forty-first International Conference on Machine Learning, 2024
> 6. Ivan Gentil, Christian Léonard, and Luigia Ripani. About the analogy between optimal transport and
> minimal entropy. In Annales de la Faculté des sciences de Toulouse: Mathématiques, volume 26,
> pp. 569–600, 2017.
> 7. Cédric Villani et al. Optimal transport: old and new, volume 338. Springer, 2009.

---

> > ### Comment · Reviewer_LqYG · 2024-11-21
> > **Response to replies of authors (part 2/2)**
> >
> > I appreciate the delicate responses of the authors. However, as for "4. Dynamic formulation" and "5. Contradiction in the Hamiltonian", I still have some concerns.
> > ___
> > **4. Dynamic formulation**
> >
> > As you pointed out in the response, the connection between the dynamic version (e.g., Schr\"odinger Bridge Problem) and the static version (e.g., EOT problem) is vital. My questions are:
> > 1. In the proof sketch, you mentioned "the optimal paths $p_1^*$ and $p_2^*$". In your proof, you discretize the path $p$ from $x_0$ to $x_1$ into $N$ "subpaths", i.e., paths $p_0, p_1, ..., p_{N-1}$. Is FSBM finding $N$ optimal subpaths? Is the discretization essential? Is integrating from $t=0$ to $t=1$ and applying Ito lemma feasible?
> >
> > ___
> > **5. Contradiction in the Hamiltonian**
> >
> > The concern is also vital because a tractable objective function is expected in OT/Bridges. My questions are:
> > 1. In the revised paper, line 1062, you use the equation $L(x,u,t)=\sup_{a} \langle a,u\rangle-H(x,a,t)$. However, the definition of convex conjugate is $H(x,a,t)=\sup_{u}\langle a,u\rangle-L(x,u,t)$. You exchange the locations of $L(x,u,t)$ and $H(x,a,t)$. Given the $\sup$, is the exchange correct? Other reviewers and AC can confirm that.
> > 2. In the training/optimization scheme of FSBM, the objective function Eq. 14 is a regression of $u_{\theta}$ over the optimal drift $u^*$. How do you obtain the target $u^*$? Is $u^*$ computed via Eq. 11 utilizing **coupled paired samples**? If yes, the optimal drift is not rigorous, because $u^*$ is computed through coupled paired samples. Strong prior information is given to the solving process of the EOT problem. If I understand the computation of $u^*$ correctly, the clustering behavior of FSBM can be interpreted. The target conditional drift is computed through coupled paired samples, thus trajectories of unpaired samples will approach trajectories of paired samples.

---

> ### Author Response · Authors · 2024-11-22
> **Author Response to Reviewer LqYG (part 1/2)**
>
> We thank the reviewer for the reply. In the following points, we address the additional concerns and questions.
>
> ### **4. Dynamic Formulation**
> - The sketch of the proof starts with only two optimal subpaths to build the intuition that it is feasible and rigorous to start from the static EOT and to add intermediate points breaking down the initial problem into subproblems., whose optimal paths can combine to yield the optimal path of the initial problem.  Subsequently, we mention in the sketch of proof that this can be extended into any number of intermediate paths i.e., $N$. As  $N\rightarrow \infty$ , this leads us to to consider ‘local’ versions of optimal transport problems between infinitesimally close distributions.
> - In our proof, we start from the EOT problem, from $t_0=0$ and $t_N=1$, directly consider $N$ distinct intermediate points at time steps $t_i$ for $i \in [0, N]$, discretizing the intermediate path into $N$ subpaths. It is highlighted that this discretization is an intermediate step to derive the dynamic expression from the static regularized EOT, by taking $N\rightarrow \infty$. This step of discretizing and considering infinitesimally many subpaths discretization is integral at recasting static OT problems into the dynamic expressions that have been widely celebrated recently [1, 2]. Therefore, all dynamic SB frameworks and formulations (including our FSBM) can be interpreted as finding $N$ optimal intermediate subpaths $p^\star_{t_i, t_{i+1}}, \ i \in [0, N-1]$ for $N\rightarrow\infty$ from the prism of dynamic EOT, as it holds that the union of these infinitesimally close and infinitesimally small optimal subpaths, is equivalent to finding the continuous optimal marginal path $p_t^\star$.
> - Finally, having obtained the dynamic formulation we can integrate between the two endpoints, i.e., from $t=0$ to $t=1$, and, given the stochastic dynamics governing the path $X_t$, any function $f(X_t)$ satisfies the Ito’s Lemma.

---

> > ### Author Response · Authors · 2024-11-22
> > **Author Response to Reviewer LqYG (part 2/2)**
> >
> > ### **5. Contradiction in the Hamiltonian**
> > - We apologize if our notation is confusing. We will provide further clarifications for the Eq. in line 1062 in the manuscript.
> > Since $L$ is a convex and continuous function, the Fenchel-Moreau theorem is satisfied, which is a sufficient condition for strong duality [3]. This suggests that at at the optimal point ($a_t^\star, u_t^\star$) , it holds that $H(x_t, a_t^\star, t) = \langle a_t^\star, u_t^\star\rangle - L(x_t, u_t^\star, t)$ [4].  Therefore, this equality can equivalently be rewritten as $L(x_t, u_t^\star, t) = \sup_a \langle a_t, u^\star_t\rangle - H(x_t, a_t, t) = \langle a_t^\star, u^\star_t\rangle - H(x_t, a^\star_t, t)$. This allows us to substitute the expression in the last line of Eq. (47) $\langle a_t^\star, u^\star_t\rangle - H(x_t, a^\star_t, t)$ with $L(x_t, u_t^\star, t)$, and express the variational gap as the Bregman Divergence.
> > - It is highlighted that the parameterized drift is matched to the conditional optimal drift $u_{t|0,1}^\star$, which is not the same as $u_t^\star$. The conditional optimal drift $u_{t|0,1}^\star$ is indeed computed from Eq. (11), but it does not correspond to the drift of the coupled paired samples from the Key Point set. The drift is computed using the uncoupled samples after the optimization in Eq. (10) from which the optimal conditional path $p_{t|0,1}^\star$ has been obtained. Here the optimality refers to the output of the optimization problem for the uncoupled pairs and not to the KPs. Matching the parameterized drift $u_t^\theta$ to the $u_{t|0,1}^\star$ which corresponds to the optimized drift of the uncoupled samples is a rigorous approach and has been the standard practice in many Bridge Matching frameworks [5, 6, 7]. In the reviewed version of the manuscript, we specify in the methodology section that the conditional probability path $p_{t|0,1}$, and $u_{t|0,1}$ refer to the uncoupled samples.
> > ---
> > ### References
> > 1. Jean-David Benamou and Yann Brenier. A computational fluid mechanics solution to the monge-kantorovich mass transfer problem. Numerische Mathematik, 84(3):375–393, 2000
> > 2. Cédric Villani et al. Optimal transport: old and new, volume 338. Springer, 2009.
> > 3. Stephen Boyd, Stephen P Boyd, and Lieven Vandenberghe. Convex optimization. Cambridge university press, 2004.
> > 4. Kirill Neklyudov, Rob Brekelmans, Daniel Severo, and Alireza Makhzani. Action matching: Learning stochastic dynamics from samples. In International conference on machine learning, pp. 25858–
> > 25889. PMLR, 2023
> > 5. Chen, T., Gu, J., Dinh, L., Theodorou, E. A., Susskind, J., and Zhai, S. Generative Modeling with Phase Stochastic Bridges. In International Conference on Learning Representations, 2024.
> > 6. Guan-Horng Liu, Yaron Lipman, Maximilian Nickel, Brian Karrer, Evangelos A Theodorou, and Ricky TQ Chen. Generalized Schrödinger bridge matching. In International Conference on Learning Representations, 2024.
> > 7. Yuyang Shi, Valentin De Bortoli, Andrew Campbell, and Arnaud Doucet. Diffusion schrödinger bridge matching, 2023.

---

> > > ### Comment · Reviewer_LqYG · 2024-11-23
> > > **Response to Authors**
> > >
> > > Thank the authors for their replies. Most of my concerns have been addressed.
> > >
> > > I still have one question about the optimal drift $u_{t|0,1}^*$. In line 8 of Algorithm 1, how do you handle the optimization of the conditional distribution $p_{t|0,1}$? As we know, Schr\"odinger Bridge Problem usually optimizes the drift term of SDE. Given the nonlinear SDE, $p_{t|0,1}$ does not have an analytical solution.

---

> ### Author Response · Authors · 2024-11-25
> **Author Response to Reviewer LqYG**
>
> We sincerely thank the reviewer for their comment and appreciate their acknowledgment that our responses have effectively addressed most of their concerns.
>
> ### **Optimization of intermediate path $p_{t|0,1}$**
> It is highlighted that we do *not* need to obtain $p_{t|0,1}$ explicitly. Conversely, for our analysis, we need to be able to:
>
> 1. efficiently sample $X_t\sim p_{t|0,1}$
> 2. compute the optimal drift from $p_{t|0,1}$: $u_{t|0,1}$ which is sufficient for computing the following matching loss.
>
> While, in general, it is correct that $p_{t|0,1}$ does not admit closed-form, but if we are willing to restrict the solution space to Gaussian paths $p_{t|0,1}\approx \mathcal{N}(I_{t|0,1}, \sigma_t \mathbf{I}d)$, as is the standard practice in state-of-the-art frameworks [1], then we can achieve both (1) and (2) very efficiently. In fact, we can easily derive a closed-form solution for the optimal conditioned drift in a simulation-free manner (see Eq. (11)). In our future work, we aim to go beyond Gaussian paths, for instance, investigate stein-variational paths for multi-modal exploration.
>
> Lastly, the iterative process to optimize with respect to the intermediate path is described in the next steps. Let $p_{t|0,1}^i $ be the initial interpolating path between the two marginals. We sample $X_t\sim p_{t|0,1}^{i}$, and evaluate the guidance function $G(X_t, x_0)$ and the conditional drift from Eq. (11). Subsequently, we evaluate the objective in Eq. (10), and take a gradient step to obtain the updated intermediate path $p_{t|0,1}^{i+1}$, and use it to sample in the next iteration. This process is repeated until convergence, where $p_{t|0,1}^\star$ is obtained. It follows that from $p_{t|0,1}^\star$, the corresponding intermediate drift is the optimal conditional drift $u_{t|0,1}^\star$ .
>
> ---
>
> ### References
> 1. Guan-Horng Liu, Yaron Lipman, Maximilian Nickel, Brian Karrer, Evangelos A Theodorou, and Ricky TQ Chen. Generalized Schrödinger bridge matching. In International Conference on Learning Representations, 2024.

---

> > ### Comment · Reviewer_LqYG · 2024-11-26
> >
> > Thanks for the responses， and I will improve my rating to 8.

---

### Official Review · Reviewer_Pfgj · 2024-11-03

**Soundness:** 3
**Presentation:** 3
**Contribution:** 3
**Rating:** 8
**Confidence:** 5

**Summary:**

Summary of the paper

The paper proposed a novel semi-supervised matching framework, termed as  Feedback Schrödinger Bridge Matching (FSBM), which was designed to improve the efficiency of distribution transport problems by incorporating a small percentage of pre-aligned pairs as state feedback. This framework was developed in order to minimalize  supervision on basis of scalability. It also addressed the limitations of existing methods, which often face trade-offs between computational efficiency and optimal pairing access during training. The authors claim that FSBM enhances generalization and accelerates training by reformulating the static Entropic Optimal Transport (EOT) problem into a dynamic one, thereby facilitating the matching of non-coupled samples.

Contributions
+ Analysis begins with a static semi-supervised OT problems and derivation of a modified dynamic formulation out of it.
+ Introduction of FSBM, the first matching framework leveraging information from partially aligned datasets and Entropic Lagrangian extension of the variational gap objective used in BM frameworks to match the parameterized drift given a prescribed probability path.
+ Verification of remarkable capability of FSBM through extensive experiments to generalize under a variety of unseen initial conditions, while simultaneously achieving faster training times. These results were consistent across a wide range of matching tasks, from low-dimensional crowd navigation to high-dimensional opinion depolarization and image translation.

Merits
+  Proposal of a semi-supervised framework. This framework leverages minimal aligned pairs, which is a significant contribution to the field, because it effectively addresses the practical challenges posed by fully unsupervised and fully supervised methods. This method is novel and efficient.
+ The paper offers a solid theoretical basis mathematically  by deriving a generalized EOT objective and demonstrating its advantages in terms of efficiency and generalization capability.
+  The extensive experiments conducted across various tasks, including crowd navigation, opinion dynamics, and image translation, provide compelling evidence of the effectiveness of FSBM compared to state-of-the-art methods. Additionally  comparative experiments and their results were also shown to compare results with methods proposed in this paper and baseline methods.
+ The inclusion of detailed performance metrics and visual comparisons strengthens the claims made, illustrating the superior performance of FSBM in practical scenarios. The results were shown vividly.

Weaknesses:
+ While the theoretical aspects are robust, the manuscript could benefit from clearer explanations and simplifications, particularly in sections discussing the derivation of dynamic formulations and coupling optimization. A few formula faults are discovered in the paper.
+ The manuscript lacks a thorough discussion regarding the limitations of FSBM, particularly in relation to scenarios with highly variable datasets or when the fraction of aligned pairs is insufficient for effective guidance.

Suggestions for Improvement:
+ Simplify the presentation of theoretical concepts to enhance readability and comprehension for a broader audience.
+ Expand the comparative analysis to include more specific scenarios where existing methods excel, providing a more nuanced view of FSBM’s performance.
+ Discuss the limitations of FSBM more explicitly, particularly regarding the trade-offs associated with the number of aligned pairs used.
+ Include suggestions for future research that could build on the findings and address any identified limitations.

**Strengths:**

See Summary

**Weaknesses:**

See Summary

**Questions:**

See Summary

---

> ### Author Response · Authors · 2024-11-20
> **Author Response to Reviewer Pfgj**
>
> We truly appreciate the encouraging comments of the reviewer, as well as their useful remarks.  In the following we address the points raised by the reviewer.
>
> ### **1. Further Simplifications and Explanations of theoretical Results**
> In the updated version of the manuscript, we have refined our explanations on key points of our framework to enhance readability. Specifically, we have elaborated on how our guidance function works in our dynamic formulation. Additionally, we have included a Figure representing the vector field of the gradient of the guidance function better capturing the effect of the guidance function and how it pushes the uncoupled data towards the keypoint trajectories. Additionally, we have clarified the importance and the role of the outputs of each optimization in the alternating optimization scheme. Lastly, we also provided a sketch of the proof of Theorem 3.2 in the Appendix to simplify our theoretical results and enhance comprehension. Lastly, the typos in the equation have been carefully detected in the paper and corrected.
>
> ### **2. Discussion of the limitations of FSBM and suggestions for future research**
> While our FSBM displayed great capabilities in distribution matching, leveraging effectively the information of partially aligned datasets to guide the transport map, we acknowledge that our framework greatly relies on the  the availability and the quality of the available KP pairs. The manner in which we sample from solution of the static SB does not ensure that all modes are equally and satisfactorily represented. This implies that in cases where a minimal number of these pairs are available, the regions of attraction formed around these may not sufficiently cover the diversity of the source distribution. This could lead to wrongful guidance, suboptimal matching and poor generalizabilty. A pertinent section discussing the limitations of our framework has been added to the paper.
>
> In future work, we have identified several directions which could improve our work even further. For instance, we aim to devise a more intricate mechanism to sample from the solution of the static SB, while guaranteeing that all modes in highly diverse datasets are covered and represented through the sampled KPs. Additionally, we aim to further explore the capabilities of the guidance function and devise more task-specific function which furhter improve perfomance generalizability by optimizing the information offered by the KP samples. Additionally, we intend to improve the way we impose guidance, by allowing the aligned pairs to preserve the relative distance to k-nearest neighboring KPs, instead of simply the nearest one.
>
> ### **3. More experimental results**
> We provided an extensive experimental comparison of our FSBM with other state-of-the-art mathcing frameworks, such as DSBM  regarding the optimality of the coupling in the generated images and the preservation of structural information.  Our FSBM is a novel method further enforcing such structures—much explicitly—via the semi-supervised guidance, and our method does lead to “better coupling” across the suggested metrics, such as L2 norm, LPIPS, and SSIM, as reported below. It is clear that FSBM outperform other SB baselines (DSBM [1] and LOSBM [2]) across almost all metrics. Note that the values of the $L2$-norm have been divided by the total number of pixels in the images (i.e., $1024\times 1024$)
>
> | $L2/(1024\times 1024)$ | W2M | M2W | O2Y | Y2O |
> | --- | --- | --- | --- | --- |
> | DSBM | 0.87 | 0.84 | 0.88 | **0.87** |
> | LOSBM | 1.42 | 1.47 | 1.41 | 1.35 |
> | FSBM | **0.79** | **0.81** | **0.83** | 0.88 |
>
> | LPIPS | W2M | M2W | O2Y | Y2O |
> | --- | --- | --- | --- | --- |
> | DSBM | 0.47 | 0.48 | 0.49 | 0.53 |
> | LOSBM | 0.70 | 0.69 | 0.71 | 0.70 |
> | FSBM | **0.44** | **0.43** | **0.47** | **0.46** |
>
> | SSIM | W2M | M2W | O2Y | Y2O |
> | --- | --- | --- | --- | --- |
> | DSBM | **0.51** | 0.46 | **0.50** | 0.43 |
> | LOSBM | 0.35 | 0.34 | 0.35 | 0.36 |
> | FSBM | 0.47 | **0.49** | 0.46 | **0.45** |
>
> ---
>
> ### References
>
> 1. Yuyang Shi, Valentin De Bortoli, Andrew Campbell, and Arnaud Doucet. Diffusion schrödinger
> bridge matching, 2023.
> 2. Nikita Gushchin, Sergei Kholkin, Evgeny Burnaev, and Alexander Korotin. Light and optimal
> schrödinger bridge matching. In Forty-first International Conference on Machine Learning, 2024

---

> > ### Comment · Reviewer_Pfgj · 2024-11-22
> > **To authors**
> >
> > After read the rebuttal about this paper, I think all the questions have been addressed, and I will update my score to 8, good luck to authors.

---

### Official Review · Reviewer_GeTG · 2024-11-03

**Soundness:** 3
**Presentation:** 3
**Contribution:** 3
**Rating:** 6
**Confidence:** 4

**Summary:**

This paper tackles a semi-paired matching for generative models, based on Schrödinger bridge. In the semi-paired matching problems, along with non-coupled data, a small portion of pre-aligned data pairs are provided. In methodology, the paper is inspired by (Gu et al., 2023a) which investigates the guided optimal transport using a few paired data in static OT setting. The authors utilize a "distance-preserving" scheme to develop the dynamic OT model with entropy regularization, i.e, the Schrödinger bridge. Building on this, the matching algorithm is developed. Experiments on three settings are conducted to evaluate the proposed methods.

**Strengths:**

- It is interesting to explore the guidance of a small fraction of paired data to data translation in flow/bridge matching.
- It is reasonable to deduce the dynamic version of keypoint-guided optimal transport with a few paired data as guidance for developing the corresponding matching models (I did not check the proof and mathematical deductions).
- The paper writing is clear with clear organization and presentation.

**Weaknesses:**

- The main weakness to me is the unclear performance gain by introducing the keypoints, from the experiments. The authors reported the W-distance or FID between generated data and real data. However, to my understanding, the tackled problem in this paper belongs to the controllable generation, in which the transported data should preserve certain structures or information of the original source data. For example, in the image-to-image translation (Woman->Man) as in the experiments of this paper, the transported images should preserve as much information as possible of the input image as long as the transported image is a "man". The authors show some transported images, but I can not find the corresponding metric. Maybe a user study is helpful for this issue, e.g., ask the GPT to evaluate the generated images. Without such a metric, the importance of the keypoints can not be illustrated (the same issue exists in the other two experiments).
- It is quite confusing that adding in more keypoints worsens the performance. Is this because the distance-preserving is not effective for such cases? The authors mentioned that Gu et al., 2022 presented the relation-preserving scheme, which seems to be more effective than the distance-preserving scheme as shown in (Gu et al., 2022). Why not consider the relation-preserving scheme instead of the distance-preserving scheme? Unfortunately, both the above issues are left as future work.
- In Eq.(7), the $i$ is selected such that $x_0^i$ is closest to $X_0$. What if selecting $i$ such that $x_1^i$ is closest to $X_1$. The authors should prove that the designed strategy is optimal or justified.

**Questions:**

- How about including the metrics, e.g., LPIPS distance, perceptual similarity loss, SSIM, $L_2$ distance, etc. between the input and output images for the image-to-image translation?
- For the user study, could you provide the generated images by different methods to distinct persons, and ask which one is better for the desired translation (e.g., better preserving the identity, background, stricture, etc.)? Meanwhile, could you ask the large language models (LLMs), e.g., GPT, to answer the question, as LLMs show a strong ability to understand text and images?
- For the other experiments, could you train a classifier on the target domain data and apply it to the generated data, to see if the accuracy could be improved with the keypoints as guidance?
- Could you try the relation-preserving scheme? At least, discuss the challenges faced. Meanwhile, could you provide more analysis on the effect of the number of keypoints in-depth, to figure out why a larger number of paired data do not bring performance gain?
- How about selecting $i$ such that $x_1^i$ is closest to $X_1$ or a weighted combination of selecting based on $X_0$ and $X_1$. Which strategy is optimal or most reasonable?

I am happy to increase the score if the questions are well addressed.

---

> ### Author Response · Authors · 2024-11-20
> **Author Response to Reviewer GeTG (part 1/3)**
>
> We would like to thank the reviewer for their positive comments, as well as their constructive criticism and further suggestions for improving our work. In the following we address the points raised by the reviewer.
>
> ### **1. Experiments to quantify information and structure preservation in transported images**
> The reviewer is correct that, for data-to-data translation tasks, we aim to preserve structures of original samples after transportation. In most prior OT methods, this is enforced—implicitly—by using a latent space that is assumed to preserve similar structures for points close to each other. Our FSBM is a novel method further enforcing such structures—much explicitly—via the semi-supervised guidance, and our method does lead to “better coupling” across the suggested metrics, such as L2 norm, LPIPS, and SSIM, as reported below. It is clear that FSBM outperform other SB baselines (DSBM [1] and LOSBM[3]) across almost all metrics. Note that the $L2$-norm is divided by the number of pixels in the images (i.e., $1024\times 1024$).
>
> | $L2/(1024\times 1024)$ | W2M | M2W | O2Y | Y2O |
> | --- | --- | --- | --- | --- |
> | DSBM | 0.87 | 0.84 | 0.88 | **0.87** |
> | LOSBM | 1.42 | 1.47 | 1.41 | 1.35 |
> | FSBM | **0.79** | **0.81** | **0.83** | 0.88 |
>
> | LPIPS | W2M | M2W | O2Y | Y2O |
> | --- | --- | --- | --- | --- |
> | DSBM | 0.47 | 0.48 | 0.49 | 0.53 |
> | LOSBM | 0.70 | 0.69 | 0.71 | 0.70 |
> | FSBM | **0.44** | **0.43** | **0.47** | **0.46** |
>
> | SSIM | W2M | M2W | O2Y | Y2O |
> | --- | --- | --- | --- | --- |
> | DSBM | **0.51** | 0.46 | **0.50** | 0.43 |
> | LOSBM | 0.35 | 0.34 | 0.35 | 0.36 |
> | FSBM | 0.47 | **0.49** | 0.46 | **0.45** |
>
> Finally, we also used ChatGPT to assess the image couplings between 100 pairs of input and generated images in each translation task. Given that the coupling from LOSBM was considerably worse than the coupling generated by the other two methods, we only performed comparison between DSBM [1] and our FSBM.  ChatGPT was instructed to give a score on the coupling quality on a scale 0-5, with 5 being the best. The coupling quality was assessed using the following 3 criteria: i) Background Consistency: consistency in terms of color, lighting, background objects, ii) Identity Preservation: maintain the identity of the original face in terms of facial features and overall resemblance  iii) Structural Integrity: preserve structural integrity of the input image in terms of accessories worn, pose, orientation.  The results in 4 translation tasks presented in the paper are shown in the Tables below.
>
> | Background Consistency | W2M | M2W | O2Y | Y2O |
> | --- | --- | --- | --- | --- |
> | DSBM | 4.05 | 4.04 | 3.99 | 4.08 |
> | FSBM | **4.58** | **4.21** | **4.05** | **4.10** |
>
> | Identity Preservation | W2M | M2W | O2Y | Y2O |
> | --- | --- | --- | --- | --- |
> | DSBM | 3.89 | 3.09 | 3.15 | 3.49 |
> | FSBM | **4.02** | **3.48** | **3.72** | **3.60** |
>
> | Structural Integrity | W2M | M2W | O2Y | Y2O |
> | --- | --- | --- | --- | --- |
> | DSBM | 3.38 | 3.27 | 3.11 | 3.30 |
> | FSBM | **3.69** | **3.61** | **3.55** | **3.33** |
>
> ### **2. Justification for selection of condition to $X_0$**
>
> We thank the reviewer for bringing up this question. While we do acknowledge that, the reason for selecting to condition on the source distribution was mainly empirically driven, we highlight that, from a mathematical perspective, our derivation could follow the same steps, and Eq. (9) would use $X_1$ as the reference point instead of $X_0$. That said, the main contribution and novelty of our proposed framework is *not* the specific instantiation of guidance functions in Eq 7, but rather a general semi-supervised matching framework for *any* guidance functions that can be problem-specific.
>
> Crucially, conditioning the guidance function on samples of either function serves as an anchor (similar to [2]) to guide the interpolation trajectories of the unpaired samples and their coupling to the opposite marginal.  That said, empirical results indicate that conditioning on the terminal distribution slightly increased the  $\mathcal{W}^2$  distance between the generated and ground-truth terminal distributions across all tasks, as detailed in the table below.
>
> |  | S-tunnel | V-neck |
> | --- | --- | --- |
> | Condition on $X_1$ | 0.07 | 0.02 |
> | Condition on $X_0$ | **0.02** | **0.01** |
>
> In future work, we also aim to enhance our guidance strategy by enabling KeyPoint pairs to maintain relative distances to k-nearest neighboring keypoints, rather than solely focusing on the nearest one. Regarding conditioning on a weighted combination of both $X_0$ and $X_1$, it is not straightforward how to impose such conditioning on the guidance function in our current mathematical formulation. That said, this is definitely an intriguing idea we had not considered before, and we would like to thank the reviewer for suggesting it.

---

> ### Author Response · Authors · 2024-11-20
> **Author Response to Reviewer GeTG (part 2/3)**
>
> ### **3. Number of key points on performance and comparison between guidance schemes.**
>
> - For our crowd-navigation experiments, both approaches were examined and found the distance-preserving scheme to be more suitable for our framework. Although the relation-preserving scheme was indeed the more effective choice in the static setting in [2], this was not the case in our applications. This could be attributed to the fact that the softmax averaging relation preserving scheme works more effectively in static frameworks. However, in our dynamic formulation, we observed empirically that the softmax averaging in relation-preserving schemes (similar to [2]) failed to generate a gradient field capable of providing effective guidance to the unpaired samples. In contrast, the distance-preserving scheme offered significantly better guidance. We have added a comparison of the gradient field generated by the relation-preserving and distance-preserving functions in the Appendix Section C of the updated manuscript.
> - In future work, we plan to investigate this phenomenon in greater detail and examine whether a combined weighted scheme, of a distance-preserving and a relation-preserving term will further enforce structural information during the transport process. It is highlighted that our main contribution lies in the derivation of a dynamic transport map introducing a more general dynamic formulation, which effectively demonstrates that we can have a semi-supervised matching framework, leveraging optimal pairings. Furthermore, it is worth noting that our analysis can be adapted to any class of differentiable regularizing functions. Identifying the best possible guidance function for every use case was beyond the scope of this work.
> - Finally, we would like to clarify the findings in Fig. 9, which is accompanied by the following table in the updated version of the manuscript.  It is noted that Figure 9 in the previous version of the paper has been moved to the Appendix Section C.1.1 and is now Figure 13. First, it is noted that our algorithm is trained to transport the samples $X_0\sim\pi_0, \text{ to  }X_1\sim \pi_1$. In the evaluation phase, we consider the following distinct tasks:
>     1. Vanilla: we draw new samples from the same distribution $\pi_0 = \mathcal{N}(\mu_0, \sigma_0)$, and try to transport them to the same target distribution $\pi_1$.
>     2. Perturbed Mean:  we draw new samples from $\pi_0^\prime$ , where $\mu_0^\prime = \mu_0 + \delta\mu$ with $\delta \mu$  being the perturbation to the mean. The goal is still to transport the unpaired samples to the same target distribution $\pi_1$.
>
>
>         | Percentage of KPs (%) |  $\mathcal{W}^2$ between $\pi_1$ and $p_1^\theta$, $x_0\sim\pi_0$  (Vanilla) |  $\mathcal{W}^2$ between $\pi_1$ and $p_1^\theta$, $x_0 \sim \pi_0^\prime$  (Perturbed Mean) |
>         | --- | --- | --- |
>         | 1 | 0.034 | 0.77 |
>         | 2.5 | 0.027 | 0.55 |
>         | 5 | 0.018 | 0.79 |
>         | 8 | 0.015 | 0.82 |
>         | 10 | 0.013 | 0.81 |
>         | 12 | 0.010 | 1.01 |
>         | 15 | 0.008 | 1.23 |
>         | 20 | 0.008 | 1.48 |
>
>  where with $p_1^\theta$ we denote the generated distribution. It is demonstrated that the vanilla performance of our framework continues to increase as the number of aligned pairs increases. However, it is shown that the empirical generalization to unseen initial conditions (e.g. when the mean of the initial distribution is perturbed) does not necessarily improve by adding more KeyPoints, suggesting an overfitting like phenomenon. Another explanation is that the hyperparameters of the framework were tuned to maximize the performance while using a small number of KeyPoints samples.  As a result, increasing the number of aligned pairs may require re-tuning the hyperparameters to maintain optimal performance. Lastly, it should be mentioned that even this decrease in generalizability is still very small compared to the benchmark methods.

---

> > ### Author Response · Authors · 2024-11-20
> > **Author Response to Reviewer GeTG (part 3/3)**
> >
> > ### **4. Use of classifier for the non-image experiments**
> > We would like to thank the reviewer for their suggestion! However, in the context of our non-image experiments, training a classifier on the target domain and applying it to the generated data to quantify distributional discrepancy between the target and the generated distribution may not be the most suitable approach for evaluation. Classifier-based evaluations are often tailored to specific tasks, and might not fully capture the nuances of distributional alignment, and efficiently compare the generated distribution between our FSBM and the benchmark methods we compared against. On the other, the metrics we currently employ are better tailored to measure distributional discrepancies. These metrics provide a more explicit evaluation of how closely the generated data aligns with the target distribution.
> >
> > Moreover, using a classifier to quantify the coupling in non-image experiments presents significant challenges. Unlike image-based tasks where features and attributes such as labels, shapes, or colors provide a natural basis for classifier training and qualitative comparison, non-image data lacks such intuitive structures. Without labels or clearly defined features, training a classifier becomes considerably harder and less interpretable. This lack of intuitive comparability diminishes the usefulness of classifier-based evaluations in these settings, making them impractical and less meaningful. Lastly, it is highlighted that there is no established metric to quantitatively assess the quality of the coupling in non-image experiments. One solution could be to compare the loss function as a means to evaluate efficient coupling, however, our FSBM demonstrates different loss function (Eq. (9)) than other matching frameworks thus making direct comparison infeasible.
> >
> > ---
> >
> > ### References
> >
> > 1. Yuyang Shi, Valentin De Bortoli, Andrew Campbell, and Arnaud Doucet. Diffusion schrödinger
> > bridge matching, 2023.
> > 2. Xiang Gu, Yucheng Yang, Wei Zeng, Jian Sun, and Zongben Xu. Keypoint-guided optimal transport,
> > 2023b
> > 3. Nikita Gushchin, Sergei Kholkin, Evgeny Burnaev, and Alexander Korotin. Light and optimal
> > schrödinger bridge matching. In Forty-first International Conference on Machine Learning, 2024

---

> ### Comment · Reviewer_GeTG · 2024-11-25
>
> Thanks for the response. My major concerns were addressed, and I am happy to increase the score for the current version. However, I still believe the only metric of distribution difference between generated data and real data does not truly reflect the importance of paired data, though it shows improved results over baselines. I suggest the authors investigate more real-world applications (e.g., biological data) or metrics to show the significance of introducing the paired data, maybe in future work.

---

### Official Review · Reviewer_T3hw · 2024-11-04

**Soundness:** 3
**Presentation:** 3
**Contribution:** 3
**Rating:** 6
**Confidence:** 1

**Summary:**

Drawing inspiration from guided optimal transport schemes, this paper introduces Feedback Schrödinger Bridge Matching, a semi-supervised matching framework that uses a small set of aligned pairs to guide the transport map of non-coupled samples.
The proposed approach is evaluated on various distribution matching tasks, such as crowd navigation, opinion depolarization, and unpaired image translation, and compared against other state-of-the-art matching frameworks.

**Strengths:**

The proposed semi-supervised approach uses only a small percentage of paired data and achieves faster training time.

Experimental validation across a diverse set of tasks demonstrates the effectiveness of the proposed approach.

**Weaknesses:**

N/A

**Questions:**

N/A

---

> ### Author Response · Authors · 2024-11-20
> **Author Response to Reviewer T3hw**
>
> We would like to thank the reviewer for valuing our work.

---

### Author Response · Authors · 2024-11-20
**Author response to all reviewers**

We thank the reviewers for their valuable insights. We are excited that the reviewers identified the importance and motivation of the problem to explore the guidance of a small fraction of paired data to data translation in flow/bridge matching (*Reviewer GeTG, Pfgj*), appreciated the novelty of our dynamic semi-supervised distribution matching framework  *(Reviewer GeTG, Pfgj, LqYG)* , acknowledged our theoretical analysis to render semi-supervised matching possible *(Reviewer LqYG, Pfgj)*, appreciated our extensive experimentation *(Reviewer LqYG, Pfgj, GeTG)* and found the paper well-written *(Reviewer GeTG).* We believe our method takes a significant step that advances modern successes of diffusion models and matching frameworks creating new avenues to leverage the information available in partially aligned datasets.

---

As *all* reviewers recognized our technical novelty, the primary criticisms (raised by Reviewers Pfgj, and LqYG) stemmed from the insufficient clarification on the presentation of the FSBM algorithm and the way the guidance function operates. We agree that these aspects might be unclearly stated in the initial submission, which led to unnecessary confusion and could mislead readers. This was never our intention. We append a revised version of our manuscript where we addressed the concerns raised by all reviewers. In the revision, we include additional clarifications on the FSBM algorithm in Sec. 3 and rearranged Sec. 4 presenting the experiments. Notable changes are marked in blue and are listed below.

1. We provide clarifications in the Methodology section, explaining what each step of the algorithm provides during the optimization process
2. We elaborate on how the guidance function works, and replaced Figure 3 with a more rigorous figure presenting how it pushes the non-paired samples to the keypoint trajectory
3. We provide a proof of sketch describing the key steps of the proof for Theorem 3.2, to enhance readibility
4. We rearranged the Expreriments section to present new results on the comparison of our FSBM with state-of-the-art matching methods in the preservation of structures of original samples after transportation. Figure 9 and the section discussing the impact of Key Point pairs in the performance of the algorithm have been moved to the Appendix Sec. C.1.1 . We also added a section discussing the limitation of FSBM and provide future research directions to further improve the current framework.

Furthermore, we carefully addressed the questions and concerns raised by all reviewers. Importantly, we provide additional experiments to present that our FSBM does lead to “better coupling” by enforcing structural information to be preserved via the semi-supervised guidance, as suggested by Reviewer GeTG. Furthermore, we also provided clarifications regarding the motivation behind the introduction of the guidance function and the way it pushes the non-coupled data to the trajectory of the keypoints; the stochastic nature of the optimal coupling; and the rigor of our dynamic formulation to Reviewer LqYG.

We hope that our new presentation has improved the manner in which our contributions in presenting a novel semi-supervised distribution matching algorithm are presented. We try our best to resolve all raised concerns in the individual responses below.

---

### Meta-Review · Area_Chair_PHrC · 2024-12-19

**Metareview:**

a) This paper considers generative modelling with semi-paired matching (in which part of the data is pre-aligned) based on Schrödinger bridge. By using a "distance-preserving" scheme, the proposed method enhances generalization and accelerates training by reformulating the static Entropic Optimal Transport (EOT) problem into a dynamic one, thereby facilitating the matching of non-coupled samples.

b) The proposed approach is the first matching framework leveraging information from partially aligned datasets. It can accelerate training and enhance generalization capabilities by using a small number of pre-aligned pairs as state feedback to guide the transfer mapping of uncoupled samples. Experiments on different tasks show that the method generalize under a variety of unseen initial conditions, while achieving faster training times.

c) The performance gain by introducing keypoints is still partially unclear. In controllable generation, the transported data should preserve structures or information of the original source data, but the initially proposed metrics (W-distance or FID) did not really consider that. As suggested by rev. GeGT, distances (L2 norm, LPIPS, and SSIM) between the original and the generated sample are measured. While this shows a correlation between the number of paired samples and the generation quality, it saturates pretty quickly without clear reason.

d) This paper proposes a novel generative approach based on Optimal Transport that can exploit some pre-matched key-points to speed-up training and improve performance. The theoretical derivation from static optimal transport to dynamic used together with an extensive validation on different generation tasks makes it an excellent paper. The weakness raised by reviewers are mostly on the experimental evaluation and are minor. Thus I recommend this paper as spot-light presentation.

**Additional Comments On Reviewer Discussion:**

Rev. T3hw could not really follow the theory in the paper so their score is not really considered.

Rev. GeTG is the most critical, pointing our problems in the evaluation metrics used and suggesting possible remedies. In addition, she/he expressed concern about the relationship between performance and number of keypoints, which is not clear. Authors followed the suggested evaluations and proposed some explanations about saturation of performance when increasing the number of keypoints. Rev. increased their score to 6.

Rev. Pfgj found the contributions of the paper important and mentioned two main limitations in the clarity of the presented theoretical formulation and the lack of a section about limitations. Authors answered to their remarks and rev. increased their score to 8.

Rev. LqYG was also quite positive and requested technical clarifications. After a few rounds of clarifications, rev. was satisfied and increased their score to 8.

Overall the paper is clearly above the acceptance threshold, with promising theoretical finding supported by empirical evidence of the quality of the approach. I propose it as a spotlight as there is still some unclarity about the relationship between performance and number of key-points.

---

### Decision · Program_Chairs · 2025-01-22

Accept (Oral)